# Quantitative proteomics and in-cell cross-linking reveal cellular reorganisation during early neuronal differentiation of SH-SY5Y cells

Marie Barth [1], Alicia Toto Nienguesso[2], Anne Navarrete Santos[2] & Carla Schmidt [1✉]

The neuroblastoma cell line SH-SY5Y is commonly employed to study neuronal function and disease. This includes cells grown under standard conditions or differentiated to neuron-like cells by administration of chemical reagents such as retinoic acid (RA) or phorbol-12-myristate-13-acetate (PMA). Even though SH-SY5Y cells are widely explored, a complete description of the resulting proteomes and cellular reorganisation during differentiation is still missing. Here, we relatively quantify the proteomes of cells grown under standard conditions and obtained from two differentiation protocols employing RA or a combination of RA and PMA. Relative quantification and KEGG pathway analysis of the proteins reveals the presence of early differentiating cells and provides a list of marker proteins for undifferentiated and differentiated cells. For characterisation of neuronal sub-types, we analyse expression of marker genes and find that RA-differentiated cells are acetylcholinergic and cholinergic, while RA/PMA-differentiated cells show high expression of acetylcholinergic and dopaminergic marker genes. In-cell cross-linking further allows capturing protein interactions in different cellular organelles. Specifically, we observe structural reorganisation upon differentiation involving regulating protein factors of the actin cytoskeleton.

[1] Interdisciplinary Research Center HALOmem, Institute of Biochemistry and Biotechnology, Charles Tanford Protein Center, Martin Luther University Halle-Wittenberg, Halle, Germany. [2] Institute of Anatomy and Cell Biology, Faculty of Medicine, Martin Luther University Halle-Wittenberg, Halle, Germany. ✉email: carla.schmidt@biochemtech.uni-halle.de

Neuronal function and dysfunction are often explored using specific model systems such as animals, neuronal stem cells or neuronal cell lines. The SH-SY5Y cell line is a well-accepted and commonly used model system, for instance, to study neurodegeneration occurring in Parkinson's or Alzheimer's disease[1]. Regulated by diverse chemical reagents, SH-SY5Y cells differentiate into specific neuronal sub-types[1–3]. As an example, retinoic acid (RA) is a vitamin A derivative inducing differentiation of SH-SY5Y cells by regulating transcription of the neurotrophin receptor, Wnt signalling and protein kinase A-dependent pathways[4–6]. Adrenergic-, dopaminergic- and predominantly cholinergic-like neuronal sub-types were identified when using RA for differentiation[1,2]. A combination of RA and phorbol-12-myristate-13-acetate (PMA) predominantly results in dopaminergic-like neurons[1,7]. Other differentiation protocols use, for instance, brain-derived neurotrophic factor (BDNF), dibutyryl cyclic adenosine monophosphate or staurosporine[8–10].

Differentiated cells express typical marker proteins of mature neurons such as synaptophysin and microtubule-associated protein 2 and, depending on the neuronal sub-type, specific neurotransmitter transporters and receptors[3,11]. Increased expression of nestin, a specific marker for early neuronal differentiation, was described after 7 days of RA treatment, followed by a decrease in expression after 16 days[12] indicating neuronal maturity after this time period.

Although many studies commonly employ undifferentiated or differentiated SH-SY5Y cells as neuronal model system, the comparative proteome analysis of the various sub-types is still incomplete[3,13]. A full proteome analysis, however, enables identification and relative quantification of neuronal marker proteins including changes in protein abundance associated with neuronal differentiation. Complementary information on protein interactions formed within the cellular environments are available from cross-linking mass spectrometry (MS). Importantly, the identification of formaldehyde cross-links by MS was recently established allowing the analysis of protein interaction networks in intact cells[14].

We set out to characterise and compare the proteomes of undifferentiated and differentiated SH-SY5Y cells. For this, we explore neuronal differentiation following standard protocols using (i) RA and (ii) a combination of RA and PMA (RA/PMA). We apply an MS-based label-free quantification approach and identify changes in the abundance of proteins, including those with a specific subcellular localisation, as well as their association with enriched KEGG pathways. Quantitative polymerase chain reaction (qPCR) further allows the characterisation of neuronal sub-types expressing specific marker genes. Employing formaldehyde cross-linking, we provide insights into protein interaction networks of undifferentiated as well as RA- and RA/PMA-differentiated cells. Our analyses increase our understanding of the proteomes of undifferentiated and differentiated SH-SY5Y cells and suggest structural rearrangements, for instance, of the actin network during neuronal differentiation.

## Results

**Proteome analysis reveals increased expression of proteins localised in the ER, plasma membrane and lysosome during neuronal differentiation**. To compare the proteomes of undifferentiated and differentiated SH-SY5Y cells, cells were first grown under standard conditions until 80% confluence. These cells were considered to be undifferentiated and represent the origin of differentiated cells. For neuronal differentiation, we followed typically employed differentiation protocols[2] including growth of undifferentiated cells at low serum conditions with RA for five days (RA-differentiation) or with RA for three days

followed by three days with RA and PMA (RA/PMA-differentiation) (see Fig. 1a and Methods section for details). Upon differentiation, the morphology of SH-SY5Y cells changed from undifferentiated cells with only few and short projections to differentiated cells with long pronounced projections (Supplementary Fig. 1). Differences between RA- and RA/PMA-differentiated cells were not recognised.

Next, we compared the proteomes of undifferentiated as well as RA- and RA/PMA-differentiated SH-SY5Y cells (see Fig. 1b for an overview on the workflow). For this, cells of 6 biological replicates of each cell culture condition were harvested, lysed and the proteins were hydrolysed using trypsin. The peptides obtained from each replicate were then analysed by liquid chromatography-coupled MS (LC-MS/MS). Following this workflow, 3661 proteins were identified after database searching in all 18 samples confirming a good reproducibility of our analyses (Supplementary Fig. 2). Accepting missing values in the replicates, 5909 different proteins were identified in total corresponding to ~60% of the expected 10,000 proteins within a cell[15]. Among the identified proteins are typical cellular proteins that occur at high copy numbers[16], for instance, actin beta, vimentin, enolase 1 as well as ribosomal proteins and histones. Proteins with low copy numbers[16] such as cadherin 2, rotatin or the mitochondrial ribosomal protein S30 were also observed (Supplementary Data 1 and 2). Importantly, we identified proteins located in different organelles confirming that our proteome analysis included all cellular compartments.

We then compared the abundance of proteins that are linked with a specific subcellular localisation in undifferentiated and differentiated cells. For this, we first calculated the relative iBAQ[17] (i.e. intensity-based absolute quantification) for each protein to determine the abundance of each protein in the individual samples. We then summed the relative iBAQ values of all proteins that are linked with the same subcellular localisation[18]. The resulting summed protein abundance of each subcellular localisation was then compared between the different cell culture conditions. In detail, we compared nine different subcellular localisations, namely the cytosol, nucleus, endoplasmic reticulum (ER), dense cytosol, mitochondrium, plasma membrane, lysosome, peroxisome and the golgi apparatus. An increase in the summed protein abundance of differentiated cells when compared with their undifferentiated origin was observed for proteins localised in the ER, plasma membrane and the lysosome (Fig. 2 and Supplementary Data 1). Notably, only minor differences were observed between RA- and RA/PMA-differentiated cells; only the total abundance of lysosomal proteins was significantly lower in RA-differentiated cells when compared with RA/PMA-differentiation. We conclude that similar changes occur in RA- and RA/PMA-differentiated cells during neuronal differentiation.

**Proteome analysis of RA- and RA/PMA-differentiated cells uncovers markers for early neuronal differentiation**. Next, we relatively quantified the proteins of the differently treated SH-SY5Y cells. For this, peptide intensities were obtained from MaxQuant[19] and normalised for all proteins in each biological replicate (see Methods section for details). A t-test was applied to identify statistically significant changes in relative protein abundance between the different growth conditions. Following this approach, ~3800 proteins were relatively quantified for pair-wise comparison. More precisely, 3787 proteins were relatively quantified when comparing RA-differentiated cells with undifferentiated cells, 3738 proteins were relatively quantified when comparing PMA/RA-differentiated cells with undifferentiated cells and 3837 proteins were relatively quantified when

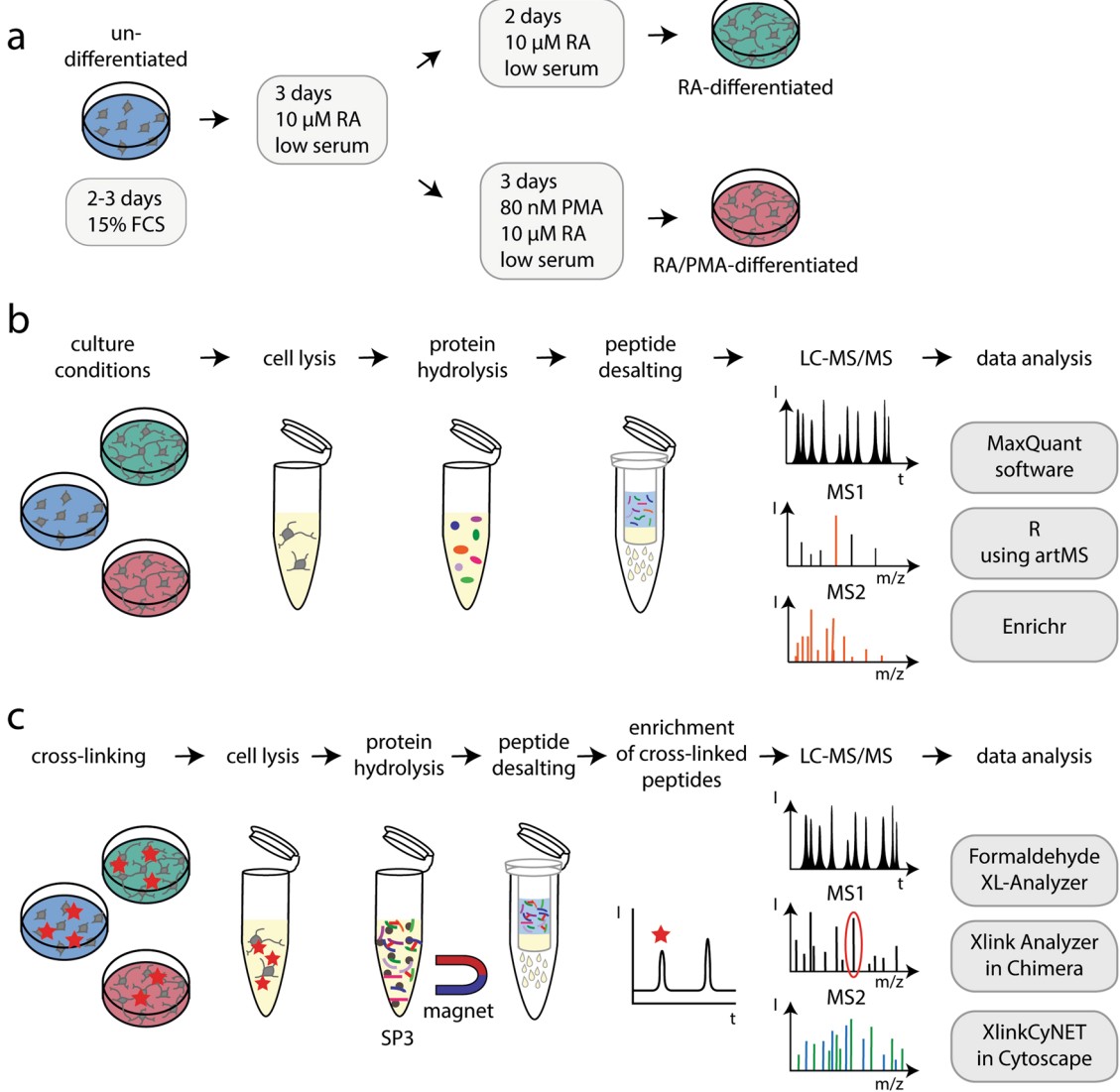

**Fig. 1 Workflow for differentiation as well as proteome and cross-linking analysis of SH-SY5Y cells. a** SH-SY5Y cells were passaged every 2–3 days and then differentiated using RA or a combination of RA and PMA (RA/PMA). See Methods section for details. **b** Workflow for proteome analysis of SH-SY5Y cells. After cell lysis, the proteins were hydrolysed and peptides were prepared for MS analysis. The peptides were analysed by LC-MS/MS followed by data analysis. See Methods section for details. **c** Proteins of SH-SY5Y cells were cross-linked with formaldehyde. After cell lysis, the proteins were hydrolysed, cross-linked peptide pairs were enriched and prepared for MS analysis followed by LC-MS/MS and data analysis. See Methods section for details.

comparing RA- and RA/PMA-differentiated cells (Supplementary Data 2). Note that a relative fold-change can only be calculated for proteins that were identified in both samples. By plotting the adjusted $p$-values against the fold-change of protein intensities, we visualised the quantitative comparison in volcano plots (Fig. 3a, b). Applying a significance threshold of 5% and a log2(fold-change) threshold of $<-0.8$ and $>0.8$, we identified 71 significantly upregulated proteins in RA-differentiated cells (Fig. 3a and Supplementary Data 2) and 101 upregulated proteins in RA/PMA-differentiated cells when compared with their undifferentiated origin (Fig. 3b and Supplementary Data 2). When comparing RA- and RA/PMA-differentiated cells, only six proteins were found to be differentially expressed (Supplementary Data 2) confirming a high similarity between the cells obtained from the two differentiation protocols.

In both, RA- and RA/PMA-differentiated cells, transglutaminase 2 (TGM2) was identified with the highest fold-change when compared with the undifferentiated cells (i.e. 16- and 13-fold

upregulation upon RA- and RA/PMA-differentiation). An increased expression of TGM2 following RA treatment was previously described[20] and therefore expected. Other proteins are also directly linked with RA treatment[21,22]. Of these, cellular retinoic binding protein was found to be highly abundant in RA-treated cells, while retinol-binding protein 1 and retinal dehydrogenase 10 were up-regulated to a lower extent (Table 1).

Importantly, we also quantified a group of proteins that play a role in neuronal development; examples are annexin A2 and cathepsin B inducing neurite outgrowth[23–25], secetrogranin II mediating neuronal differentiation[26], vasodilator stimulated phosphoprotein playing a role in filipodia formation[27], doublecortin like kinase 1 mediating structural rearrangements[28] and sequestosome 1 regulating the metabolic shift from aerobic glycolysis to oxidative phosphorylation[29] (Table 1 and Supplementary Data 2). The expression of nestin, a marker protein for early neuronal differentiation[12], was found to be increased in differentiated cells (Table 1), while doublecortin, a marker for

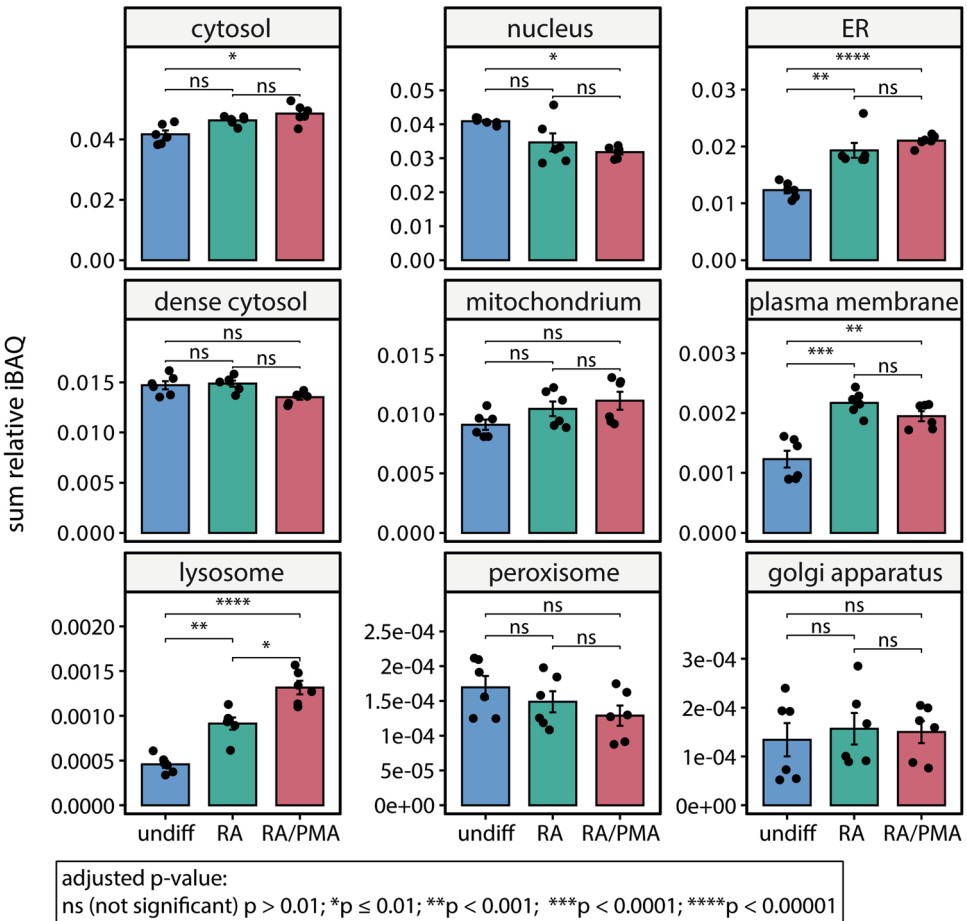

**Fig. 2 Relative quantification of proteins that are linked with a specific subcellular localisation.** The relative iBAQ value of all proteins that are linked with a specific subcellular localisation identified in undifferentiated (undiff, blue), RA-differentiated (RA, green), or RA/PMA-differentiated (RA/PMA, red) cells was summed. For each cell culture condition, the mean value and the standard error are given ($n = 6$). The adjusted *p*-value was calculated using a two-tailed *t*-test employing the Bonferroni correction method and used to determine significant differences between the conditions (see legend for details). Figure generated using Supplementary Data 1.

developing neurons[30], was down-regulated. Proteins related with the synaptic vesicle cycle (e.g. synaptophysin, SNAP25 and synaptotagmin 11) were only quantified in differentiated cells while they were absent in undifferentiated SH-SY5Y cells (Supplementary Data 2). The changes in protein expression observed upon differentiation suggest that neuron-like cells, which already established a neuronal network through synapses, are obtained. Downregulation of doublecortin reveals that very early differentiation stages have passed.

In addition, proteins related with antioxidant defence were found to be upregulated upon differentiation (Table 1). Neurons have a high energy demand and oxygen consumption and are particularly vulnerable to oxidative stress[31]. Upregulation of these proteins, therefore, indicates that an antioxidant defence mechanism is already established during early differentiation. In summary, treatment with RA and RA/PMA leads to the expression of proteins related to RA administration, neuronal development, antioxidant defence mechanism as well as synaptic proteins.

**KEGG pathway analysis reveals structural rearrangements during neuronal differentiation.** We next identified enriched KEGG pathways by analysing protein groups that were significantly enriched in differentiated SH-SY5Y cells when compared with their undifferentiated sources. For this, the genes of upregulated proteins in differentiated cells were used as input for

the gene set search engine Enrichr[32–34]. We identified 21 and 20 enriched KEGG pathways for RA- and RA/PMA-differentiated cells, respectively (Supplementary Data 2). Among the top six enriched KEGG pathways in both differentiated cells are the 'regulation of actin cytoskeleton', 'extracellular matrix receptor (ECM-receptor) interaction' and 'focal adhesion' pathways (Fig. 3c, d) which are closely related to structural rearrangements of the cells. Accordingly, the observed long projections of differentiated SH-SY5Y cells (Supplementary Fig. 1) likely require increased focal adhesion and ECM interactions. Of these pathways, proteins such as the vasodilator stimulated phosphoprotein, promoting filopodia formation, or paxillin, playing a role in F-actin assembly and dynamics[35–37], were identified in differentiated cells.

The 'riboflavin metabolism' pathway was specifically enriched in RA-differentiated cells and was assigned with the highest score. Two proteins, namely biliverdin reductase B and acid phosphatase 2, are upregulated; these enzymes catalyse reduction of flavin mononucleotide to flavohydroquinone. This catalytic activity might be relevant for oxygen activation during differentiation[38].

The pathway 'lysosome' was highly enriched in RA/PMA-differentiated cells as well as in RA-differentiated cells, albeit with a lower score. The lysosome and the corresponding proteins are required for macromolecular degradation of proteins, lipids, DNA, RNA and carbohydrates, thereby, enabling recycling of the obtained components. Important roles of axonal and dendritic

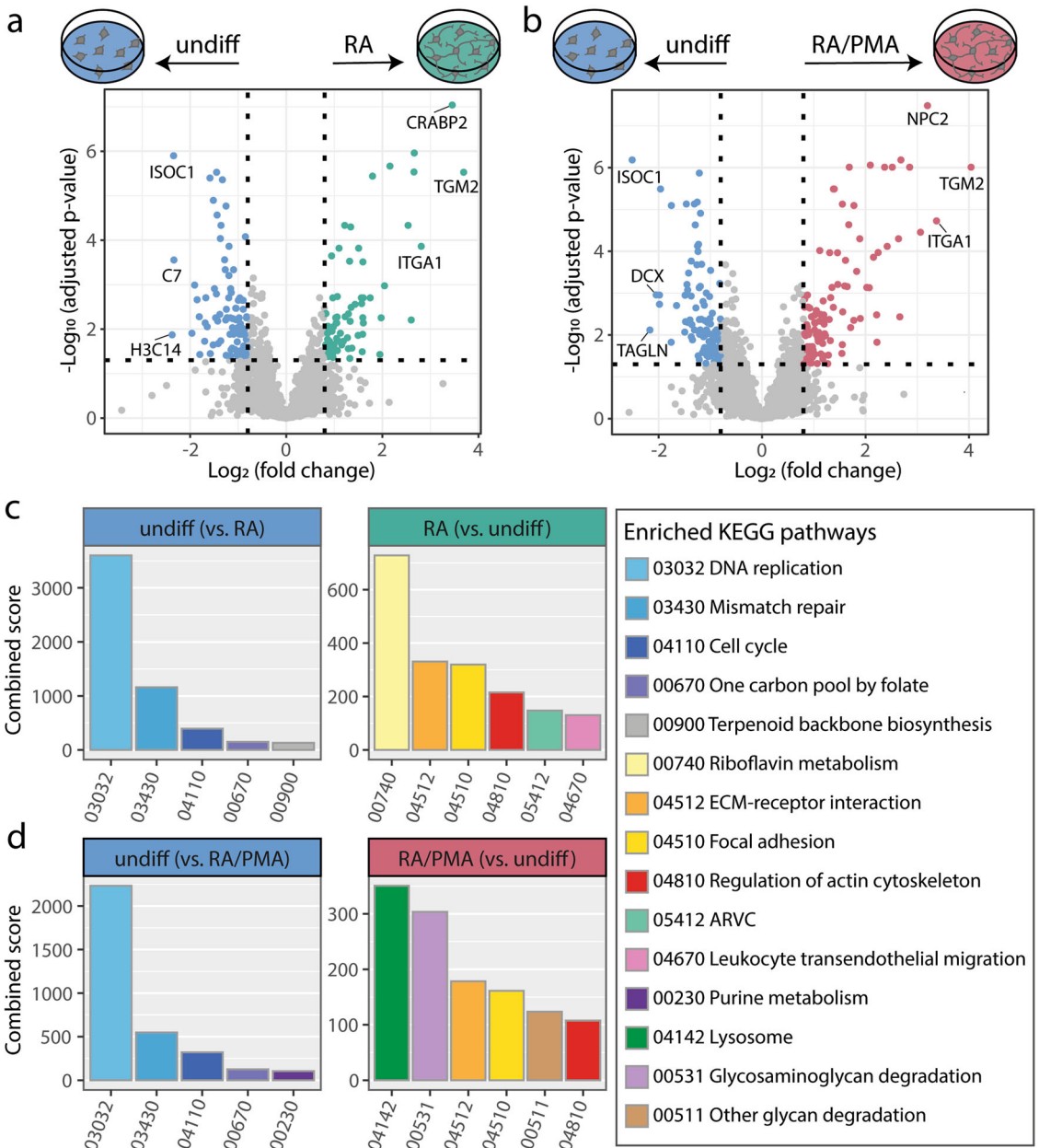

**Fig. 3 Relative quantification and enriched KEGG pathway analysis of undifferentiated versus RA- and RA/PMA-differentiated SH-SY5Y cells.** For relative global protein abundance quantification of the different cell culture conditions, a two-tailed t-test was performed and adjusted p-values were calculated according to Benjamini and Hochberg. The log2(fold-change) was plotted against the −log10(adjusted p-value) in volcano plots. Quantified proteins with a log2(fold-change) <−0.8 or >0.8 and an adjusted p-value < 0.05 were considered significantly up- or down-regulated, respectively. The gene names of the three top protein hits up- or downregulated in undifferentiated or differentiated cells are labelled in the volcano plots. Significantly up- or downregulated proteins were used for KEGG pathway enrichment analysis. **a** Volcano plot of RA-differentiated cells (RA, green) versus undifferentiated cells (undiff, blue). **b** Volcano plot of RA/PMA differentiated cells (RA/PMA, red) versus undifferentiated cells (undiff, blue). **c** Enriched KEGG pathways with high combined scores in RA-differentiated cells vs. undifferentiated cells (see legend for enriched pathways). **d** Enriched KEGG pathways with high combined score in RA/PMA differentiated vs. undifferentiated cells (see legend for enriched pathways). ISOC1 isochorismatase domain containing 1, C7 complement C7, H3C14 H3 clustered histone 14, ITGA1 integrin subunit alpha 1, CRABP2 cellular retinoic acid-binding protein 2, TGM2 transglutaminase 2, DCX doublecortin, TAGLN transgelin, NPC2 NPC intracellular cholesterol transporter 2, ECM extra cellular matrix, ARVC arrhythmogenic right ventricular cardiomyopathy. Figure generated using Supplementary Data 2.

lysosomes are clearance of dysfunctional synaptic proteins as well as aged or damaged proteins and organelles[39–41], and several genetic disorders have been linked to lysosomal dysfunction in neurons[42]. Enrichment of the KEGG pathway 'lysosome' in differentiating SH-SY5Y cells, therefore, confirms that these cells have a higher demand on degradation processes.

The KEGG pathway analysis of proteins identified in differentiated cells, again, highlights the similarity between the two differentiated sub-types. Importantly, enriched KEGG pathways are associated with structural and functional changes of the cells during neuronal development including, for instance, actin regulation and ECM interaction. In agreement with the

**Table 1 Changes in protein expression upon differentiation.**

| Uniprot accession number | Gene name | Protein name | RA vs. undifferentiated | | | RA/PMA vs. undifferentiated | | |
|---|---|---|---|---|---|---|---|---|
| | | | FC | log2FC | adj. *p*-value | FC | log2FC | adj. *p*-value |
| RA-treatment related | | | | | | | | |
| P21980 | TGM2 | Transglutaminase 2 | 12.93 | 3.69 | 2.97E-06 | 16.46 | 4.04 | 9.78E-07 |
| P29373 | CRABP2 | Cellular retinoic acid binding protein 2 | 10.97 | 3.46 | 9.13E-08 | 5.73 | 2.52 | 9.78E-07 |
| P09455 | RBP1 | Retinol-binding protein 1 | 2.45 | 1.29 | 4.53E-03 | 2.13 | 1.09 | 9.59E-03 |
| Q8IZV5 | RDH10 | Retinol dehydrogenase 10 | 3.01 | 1.59 | 1.30E-02 | 3.73 | 1.9 | 4.04E-03 |
| Neuronal differentiation | | | | | | | | |
| P07355 | ANXA2 | Annexin A2 | 1.99 | 0.99 | 2.23E-02 | 2.3 | 1.2 | 5.49E-03 |
| P07858 | CTSB | Cathepsin B | 2.7 | 1.43 | 2.77E-03 | 3.13 | 1.65 | 6.96E-04 |
| P48681 | NES | Nestin | 2.09 | 1.06 | 1.85E-03 | 3.42 | 1.78 | 8.06E-06 |
| P13521 | SCG2 | Secretogranin II | 3.93 | 1.98 | 5.56E-03 | 2.93 | 1.55 | 1.70E-02 |
| P50552 | VASP | Vasodilator-stimulated phosphoprotein | 2.09 | 1.06 | 3.03E-02 | 2.34 | 1.23 | 1.04E-02 |
| Q13501 | SQSTM1 | Sequestosome 1 | 5.8 | 2.54 | 4.64E-05 | 4.74 | 2.24 | 1.07E-04 |
| O15075 | DCLK1 | Doublecortin like kinase 1 | 1.85 | 0.89 | 2.39E-02 | 2.28 | 1.19 | 3.31E-03 |
| Oxidative stress related | | | | | | | | |
| O75874 | IDH1 | Isocitrate dehydrogenase (NADP(+)) 1 | 1.83 | 0.87 | 1.02E-02 | 1.8 | 0.85 | 9.33E-03 |
| P04424 | ASL | Argininosuccinate lyase | 3.02 | 1.59 | 5.53E-03 | 4.21 | 2.07 | 7.41E-04 |
| P30043 | BLVRB | Biliverdin reductase B | 2.55 | 1.35 | 3.24E-03 | 3.02 | 1.6 | 6.74E-04 |
| P02795 | MT2A | Metallothionein 2A | 3.86 | 1.95 | 3.72E-02 | 4.66 | 2.22 | 1.49E-02 |
| P15559 | NQO1 | NAD(P)H quinone dehydrogenase 1 | 1.94 | 0.95 | 1.11E-02 | 2.19 | 1.13 | 3.23E-03 |
| P07602 | PSAP | Prosaposin | 1.82 | 0.87 | 8.59E-03 | 3.19 | 1.68 | 2.32E-05 |
| P04179 | SOD2 | Superoxide dismutase 2 | 2.99 | 1.58 | 3.23E-02 | 2.91 | 1.54 | 2.76E-02 |
| P13521 | SCG2 | Secretogranin II | 3.93 | 1.98 | 5.56E-03 | 2.93 | 1.55 | 1.70E-02 |
| Proliferation | | | | | | | | |
| Q96CN7 | ISOC1 | Isochorismatase domain containing 1 | 0.2 | −2.34 | 1.26E-06 | 0.18 | −2.51 | 6.56E-07 |
| P49736 | MCM2 | Minichromosome maintenance complex component 2 | 0.42 | −1.25 | 1.71E-05 | 0.4 | −1.31 | 7.41E-06 |
| P25205 | MCM3 | Minichromosome maintenance complex component 3 | 0.44 | −1.2 | 1.98E-03 | 0.37 | −1.42 | 3.31E-04 |
| P33991 | MCM4 | Minichromosome maintenance complex component 4 | 0.44 | −1.19 | 6.22E-04 | 0.47 | −1.08 | 9.95E-04 |
| P33992 | MCM5 | Minichromosome maintenance complex component 5 | 0.45 | −1.14 | 1.51E-03 | 0.44 | −1.19 | 6.89E-04 |
| Q14566 | MCM6 | Minichromosome maintenance complex component 6 | 0.39 | −1.36 | 4.64E-05 | 0.42 | −1.25 | 7.46E-05 |
| P33993 | MCM7 | Minichromosome maintenance complex component 7 | 0.44 | −1.19 | 1.37E-04 | 0.43 | −1.23 | 6.81E-05 |
| P06493 | CDK1 | Cyclin dependent kinase 1 | 0.37 | −1.45 | 2.97E-06 | 0.44 | −1.19 | 1.23E-05 |
| O43602 | DCX | Doublecortin | 0.28 | −1.85 | 2.18E-03 | 0.24 | −2.05 | 1.10E-03 |
| P00374 | DHFR | Dihydrofolate reductase | 0.35 | −1.52 | 1.27E-05 | 0.36 | −1.46 | 7.41E-06 |
| P04818 | TYMS | Thymidylate synthetase | 0.37 | −1.43 | 6.56E-03 | 0.5 | −1.01 | 2.57E-02 |

The Uniprot accession number, the gene and protein name, the fold-change (FC), the log2(FC) and the adjusted p-value (adj. *p*-value) are given for selected protein groups. Fold-change ratios were calculated for RA-differentiated versus undifferentiated cells as well as for RA/PMA-differentiated versus undifferentiated cells.

quantification of proteins associated with specific organelles, only minor differences concerning lysosomal proteins were observed upon RA- and RA/PMA-differentiation.

**Proliferation is a main characteristic of undifferentiated SH-SY5Y cells.** Applying the same thresholds as described above, we identified 86 and 98 proteins that were significantly upregulated in undifferentiated cells when compared with RA- and RA/PMA-differentiated cells, respectively (Fig. 3a, b and Supplementary Data 2). One of the highest upregulated proteins is isochorismatase domain-containing protein 1 (Table 1), which was recently related to ontogenesis and knockdown in pancreatic cancer induced apoptosis as well as suppressed cell proliferation[43]. Actin beta like 2, an actin isoform, which was found to be upregulated in colorectal cancer[44], was also enriched (Supplementary Data 2). Moreover, proteins regulating cell cycle

checkpoints, such as cyclin-dependent kinase 1[45] and mini-chromosome maintenance proteins[46], are enriched in undifferentiated cells (Table 1). The upregulation of cancer-related as well as cell-cycle proteins confirms the cancer characteristics of undifferentiated SH-SY5Y neuroblastoma cells.

As described above for differentiated cells, we also identified relevant upregulated KEGG pathways of undifferentiated SH-SY5Y cells. These pathways included 'DNA replication', 'mismatch repair cell cycle' and 'one carbon pool by folate' once again demonstrating high proliferation levels (Fig. 3c, e and Supplementary Data 2). Representatives of the 'DNA replication' pathway are, for instance, the minichromosome maintenance proteins 2 to 7 acting as DNA helicases and required during cell cycle[46–48]. MutS homologs 2 and 6 were assigned to the 'mismatch repair pathway' recognising mispaired bases or insertion loops[49–51]. A decrease in the expression of these two proteins upon differentiation was

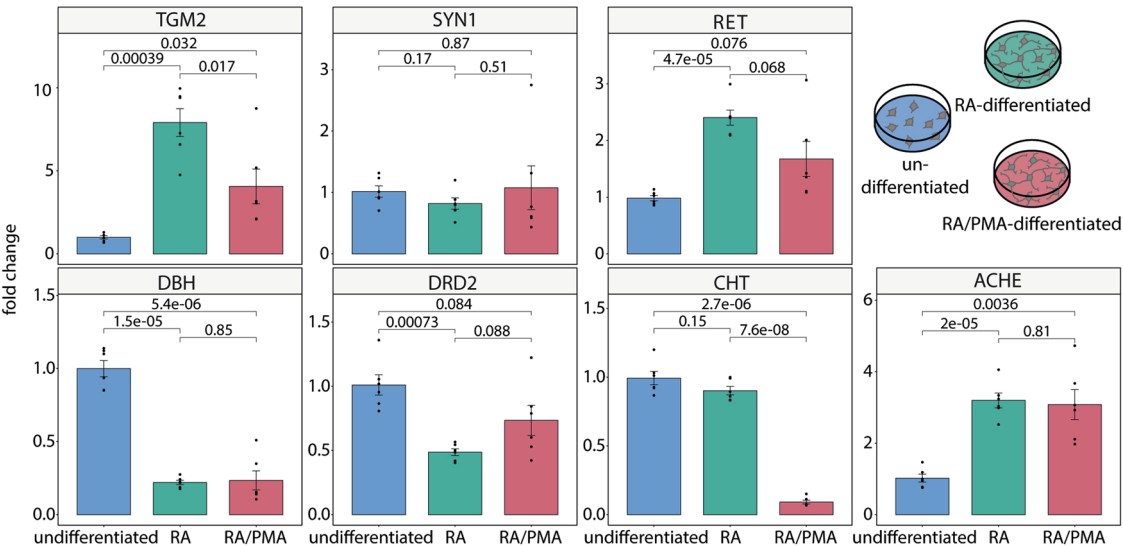

**Fig. 4 Relative expression of marker genes to determine neuronal sub-types.** qPCR was used to determine the expression of neuronal marker genes ($n = 6$). The following marker genes were analysed in undifferentiated (blue), RA-differentiated (green) and RA/PMA-differentiated (red) cells: transglutaminase 2 (*TGM2*), synapsin 1 (*SYN1*), ret proto-oncogene (*RET*), dopamine beta hydroxylase (*DBH*), dopamine receptor D2 (*DRD2*), choline transporter (*CHT*), and acetylcholin esterase (*ACHE*). Specific gene amplification was normalised to hypoxanthine phosphoribosyl-transferase 1 (*HPRT1*). The relative expression of the marker genes was calculated as fold- change of the RNA amounts in RA and RA/PMA relative to the RNA amount in undifferentiated cells. The $2^{-\Delta\Delta Ct}$ method was used. Error bars represent the standard error. A two-tailed *t*-test between two conditions was used for calculation of *p*-values. Figure generated using Supplementary Data 3.

previously reported[52]. The 'one carbon metabolism' plays an important role in maintaining high proliferation rates of cancer cells by providing one-carbon units for biosynthesis and redox reactions. Two proteins of this pathway, namely dihydrofolate reductase and thymidylate synthase, were upregulated (Table 1). When comparing the enriched KEGG pathways of undifferentiated cells with the pathways of differentiated cells, typical cancer characteristics are observed. Proteins identified in the respective pathways are commonly required for constant proliferation of the cells and cancer progression. We conclude that undifferentiated SH-SY5Y cells represent neuroblastoma cells which do not show a typical neuronal character.

**Characterisation of neuronal sub-types of SH-SY5Y cells.** The observed changes in protein expression upon differentiation of SH-SY5Y cells indicate differentiation towards mature neurons. However, specific markers for neuronal sub-types were not identified in the proteome analysis and we, therefore, investigated the expression of specific marker genes in detail. For this, RNA was isolated, transcribed into cDNA and expression levels were determined by qPCR. The $2^{-\Delta\Delta Ct}$-method[53] was then used to calculate the relative fold gene expression levels using hypoxanthine phosphoribosyl-transferase 1 (*HPRT1*) as reference gene. mRNA and protein expression have been reported to correlate for many proteins[17]. Accordingly, *TGM2* was found to be upregulated in differentiated cells by relative protein quantification and qPCR (Fig. 4 and Supplementary Data 3). The neuronal marker protein synapsin 1 (*SYN1*), on the other hand, showed similar mRNA expression levels in all three cell culture conditions. mRNA levels of ret proto-oncogene (*RET*), which is typically observed in dopaminergic neurons mediating differentiation, branching and neurite outgrowth[54], were significantly higher in RA-differentiated cells than in RA/PMA-differentiated or undifferentiated cells. Positive transcriptional regulation of *RET* was previously described to be associated with early neuronal differentiation of SH-SY5Y cells[55] indicating that early differentiation

stages have passed when following the RA/PMA-differentiation protocol.

We also targeted several neurotransmitter transporters and receptors as well as related enzymes. Of these, dopamine beta-monooxygenase (*DBH*) converts dopamine to the neurotransmitter norepinephrine (noradrenaline) and is, therefore, a marker for noradrenergic neurons. The dopamine D2 receptor (*DRD2*), the choline transporter (*CHT*) and acetylcholine esterase (*ACHE*) are markers for dopaminergic, cholinergic and acetylcholinergic neurons, respectively. *DBH* and *DRD2* showed significantly higher expression levels in undifferentiated cells when compared with RA- and RA/PMA-differentiated cells (*DBH*) or RA-differentiated (*DRD2*) cells. *CHT* expression was significantly higher in undifferentiated and RA-differentiated cells compared to RA/PMA-differentiated cells. Upregulation of *ACHE* was observed in both, RA- and RA/PMA-differentiated cells.

In a nutshell, downregulation of *DBH* and *DRD2* and upregulation of *CHT* and *ACHE* in RA-differentiated cells suggest that cholinergic and acetylcholinergic neuronal sub-types are established. The expression of *DBH* and *CHT* was downregulated in RA/PMA-differentiated cells, while *DRD2* was expressed at similar levels and *ACHE* was upregulated when compared with undifferentiated cells (Fig. 4 and Supplementary Data 3). Accordingly, RA/PMA-differentiated cells show characteristics of dopaminergic and acetylcholinergic neurons. We conclude that undifferentiated SH-SY5Y cells do not show a defined neuronal sub-type, while RA-differentiated cells are cholinergic and acetylcholinergic and RA/PMA-differentiated cells express markers of dopaminergic and acetylcholinergic neurons.

**In-cell cross-linking uncovers protein interactions in undifferentiated and differentiated SH-SY5Y cells.** To explore the structural differences between undifferentiated and differentiated SH-SY5Y cells observed by quantitative proteomics, we next performed in-cell cross-linking. For this, undifferentiated as well as RA- and RA/PMA-differentiated cells were incubated with

formaldehyde and proteins in close proximity were covalently cross-linked. After cell-lysis, proteins were hydrolysed with trypsin and low-abundant cross-linked peptide pairs were enriched by size exclusion chromatography. The corresponding peptide fractions were subsequently analysed by LC-MS/MS and cross-linked peptide-pairs were identified using the 'Formaldehyde XL Analyzer'[14] (Fig. 1c). To increase reliability of cross-link identification, only cross-links that were identified with at least intermediate confidence in three replicates or with high and intermediate confidence in at least two replicates were accepted for further analysis (see Methods section for details). For estimation of the false discovery rate, a decoy database search was included in the analysis. Only one cross-linked peptide pair containing one peptide with a decoy sequence was identified in RA/PMA-differentiated cells, confirming a low rate of false positive hits among the identified cross-links in all cell culture conditions. Applying these confidence thresholds, 125 inter-protein cross-links and 363 intra-protein cross-links were identified in undifferentiated cells (Supplementary Fig. 3 and Supplementary Data 4). In RA- and RA/PMA-differentiated cells, 229 and 204 inter-protein cross-links as well as 713 and 689 intra-protein cross-links were identified, respectively (Supplementary Data 4).

The stringent analysis thresholds applied here resulted in fragment spectra of high quality. An example fragment spectrum of a cross-linked peptide pair is shown in Fig. 5a. Both peptide sequences, originating from histones H4 and H2B, were identified by a series of y-ions. In addition, a series of y-ions shifted in mass by 12 Da was observed. Formaldehyde primarily cross-links lysine and arginine residues but also asparagine, histidine, aspartic acid, tyrosine and glutamine residues[14]. As the linkages introduced by formaldehyde are cleaved during peptide fragmentation, it is often impossible to determine the exact cross-linking site within the peptide sequence. We, therefore, considered those amino acids located in the middle of each cross-linked peptide sequence as cross-linking site. The ions corresponding to the 12 Da-mass shift include the linker introduced by formaldehyde and, in some cases, help assigning the cross-linking site. Accordingly, the cross-linking site of the first peptide sequence (H4: TVTAMDV-VYAL(R)K) is identified by the missing mass shift for the y1-ion. Due to missing fragment ions, the cross-linking site of the second peptide (H2B: IAGEASRLAHYNK) cannot unambiguously be assigned. However, the observed 12 Da-mass shift of the y9 ion and the high reactivity of arginine residues suggest covalent linkage of the arginine residue.

Interaction networks of binary protein interactions were then generated for all three cell culture conditions (Supplementary Figs. 4–6). Due to the membrane permeability of formaldehyde, proteins of different cellular compartments were covalently cross-linked. These include histones located in the nucleus, several mitochondrial proteins, cytosolic chaperons and cytoskeletal proteins. High abundant proteins, such as histones, ribosomal proteins and actin-related proteins represent the major interaction partners.

**Identified cross-links satisfy distance restraints in available high-resolution structures**. To structurally validate our cross-linking approach, we visualised the identified cross-links on available high-resolution structures. For this, we first chose a model structure of nucleosome core particles containing two copies each of histones H2A, H2B, H3 and H4[56]. Measuring the distances between the assigned cross-linking sites and allowing for flexibility of amino acid side chains shows that the majority of cross-links identified in all cell culture conditions satisfies a distance restraint <30 Å (Fig. 5b–d). Only one (RA/PMA-differentiated) or two (undifferentiated and RA-differentiated) cross-

links above this distance restraint were identified. Notably, these cross-links originate from two copies of the same histones, as suggested by overlapping peptide sequences (Supplementary Data 4), and most likely reflect interactions between two different nucleosome particles.

Following validation of our approach using a small and stable protein complex, we next inspected a large and flexible protein machinery. One such machinery is the ribosome, which assembles from two subunits each containing a multitude of proteins and RNA molecules, and requires large structural flexibility for its function. Again, identified cross-links were visualised on an available high-resolution structure of the ribosome[57] (Fig. 6, Supplementary Fig. 7 and Supplementary Data 4). We first visualised the cross-links of ribosomal proteins identified in undifferentiated cells. 43 cross-links with a Cα distance <30 Å and one cross-link with a Cα distance >30 Å were observed (Fig. 6). Similar results were obtained for cross-links of differentiated cells (Supplementary Fig. 7). Interestingly, for two cell culture conditions, the same over-length cross-link between protein subunits RS3 (lysine 60) and RL11 (threonine 74) was observed, suggesting flexibility of the respective region in the ribosomal complex. Indeed, considering the sequence of mRNA translation, large structural rearrangements of the small ribosomal subunit are required for translocation of the ribosome on the mRNA likely accounting for the observed over-length cross-link. Additional over-length cross-links were observed in similar regions of the ribosome (Supplementary Fig. 7) suggesting that different structural snapshots of the active ribosome are captured. The structural analysis of identified cross-links, therefore, validates our methodological approach and further provides insights into structural flexibility, which is not attainable from static structures, representing snapshots of functionally active protein machines.

**Actin reorganisation in early differentiating cells**. Actin gamma 1 (an isoform of γ-actin) is among the high abundant proteins in SH-SY5Y cells (Supplementary Data 1). It is a key cytoskeletal protein and important for the development of the distinct cell shape of neurons[58]. Our proteome analysis revealed enrichment of the KEGG pathway 'regulation of actin' in differentiated cells. We, therefore, inspected the interaction network of actin gamma 1 closely. Actin gamma 1 was found to be cross-linked to nine, twelve and 15 other proteins in undifferentiated, RA-differentiated or RA/PMA-differentiated cells, respectively (Fig. 7a–c). Accordingly, cross-links between actin gamma 1 and tropomyosin as well as transgelin-2, drebrin and actin α cardiac muscle (an isoform of α-actin) were observed in all three cell culture conditions (Fig. 7a). These interactions were described previously as follows: α- and γ-actin polymerise into filamentous actin (F-actin) in vitro[59] while tropomyosin stabilizes F-actin[60] and drebrin modifies the double helix structure comprising two F-actin strands[61]. Interactions of transgelin-2 with actin were also described[62]. We identified cross-links between actin gamma 1 and the C-terminal peptide of transgelin-2 as well as its N-terminal region proximal to the calponin homology 3 (CH3) domain, both representing known actin-binding sites (Fig. 7b, c)[63,64]. Additional cross-links between the N-terminal part of transgelin-2 and tropomyosin were also observed. Due to sequence similarity of transgelin-2 with calponin, a tropomyosin binding site within its CH3 domain was predicted[65]. Our cross-linking experiments experimentally confirm this interaction.

In addition to these interactions, differentiation-specific cross-links between actin and an actin-dynamic regulating protein named adenylate cyclase-associated protein 1 were observed in differentiated cells. This protein catalyses nucleotide exchange of

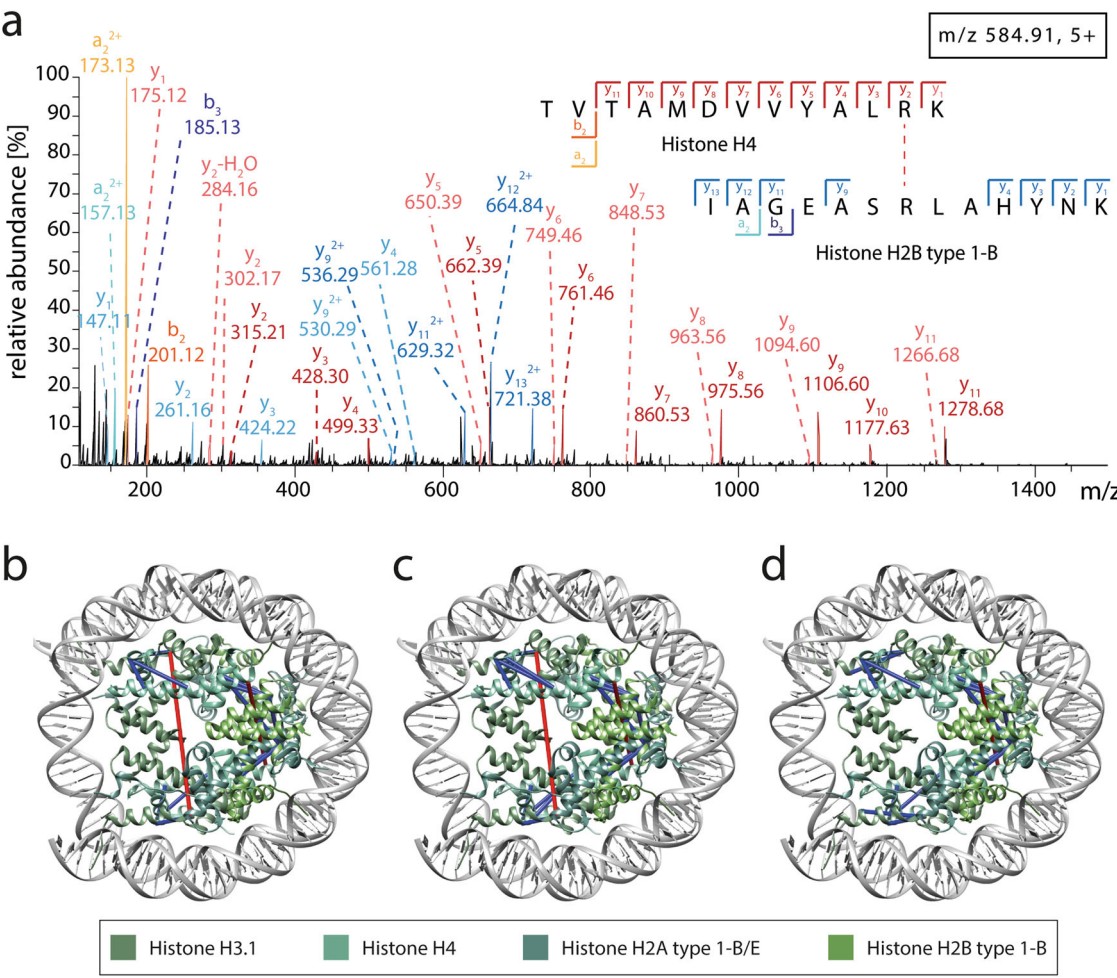

**Fig. 5 Cross-links identified for histones. a** Example fragment spectrum of an inter-protein cross-link between histone H4 and histone H2B. The m/z and charge state of the precursor ion are given. Y-ions (light red), y-ions including the mass shift of 12 Da (dark red), a-ions (orange) and b-ions (yellow) of the peptide originating from histone H4 as well as y-ions (light blue), y-ions including the mass shift of 12 Da (blue), b-ions (dark blue) and a-ions (cyan) of the peptide originating from histone H2B type 1-L are assigned. Note that not all observed fragment ions are annotated. **b–d** High-resolution structure of the nucleosome core particles containing two copies of histones H3, H4, H2A and H2B (PDB ID: 3x1s[56]) are shown. Identified cross-links are visualised in the structure (red and blue lines). Cross-linking distances >30 Å (red) and <30 Å (blue) are indicated. **b** Cross-links identified in RA-differentiated cells. **c** Cross-links identified in RA/PMA-differentiated cells. **d** Cross-links identified in undifferentiated cells. Figure generated using Supplementary Data 4.

ADP to ATP upon association with two globular actin monomers[66] and presumably accelerates actin reorganisation during early neuronal differentiation[67]. Interactions between adenylate cyclase-associated protein 1 and actin gamma 1, therefore, suggest reorganisation and further establishment of the cytoskeleton.

In RA-differentiated cells, specific interactions between actin gamma 1 and the actin-related protein 2/3 (ARP 2/3) complex, a key nucleator of actin branches[61], as well as profilin-1, a promotor of F-actin assembly[68,69], were identified (Fig. 7b). Actin filament branch formation is required for the development of dendritic networks suggesting that dendrites are established during differentiation. Interactions with laminin B were also specific to RA-differentiated cells implying co-localisation of laminin B2 mRNA and actin filaments for mRNA translation on the cytoskeleton as described previously[70].

Specific interactions in RA/PMA-differentiated cells were observed between actin gamma 1 and actin-dependent molecular motor proteins (termed myosins)[71]. Both, myosin 9 and myosin 10, are isoforms of non-muscle myosin II selectively expressed in the nervous system; they facilitate cell morphological and regulate actin dynamics[72]. In RA/PMA differentiated cells, these myosin isoforms likely influence regulation of actin. In summary, we identified changes in interactions of actin gamma 1 with actin-regulating proteins, such as the ARP2/3 complex in RA- or myosin variants in RA/PMA-differentiated cells.

## Discussion

We quantitatively compared the proteomes of undifferentiated as well as RA- and RA/PMA-differentiated SH-SY5Y cells. Relative protein abundances and KEGG pathway analyses suggest structural reorganisation of the cells during early neuronal differentiation. Interestingly, the two differentiated cell-types were remarkably similar. Protein abundances and enriched KEGG pathways of undifferentiated cells, on the contrary, suggest a rather cancer-related character with high proliferation levels. As specific marker proteins of neuronal sub-types were not identified in our proteome analysis, qPCR was employed to characterise the sub-types of the different cells. In-cell cross-linking then revealed protein interactions of cytosolic, nuclear, mitochondrial and other compartmental proteins. The pronounced and branched actin-network of differentiated cells further confirmed structural rearrangements during neuronal differentiation in agreement with our proteome analysis.

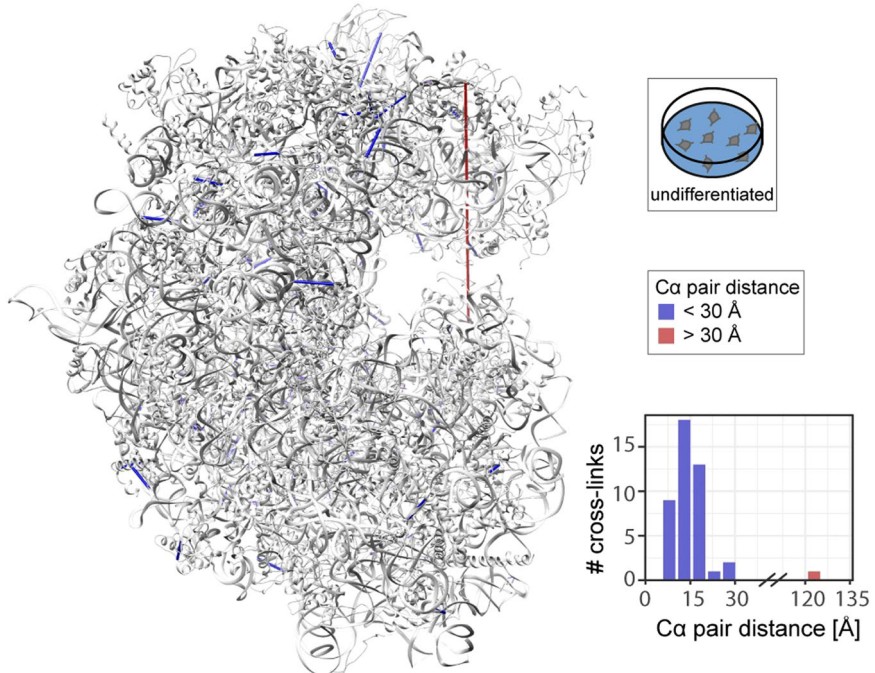

**Fig. 6 Cross-links of ribosomal proteins identified in undifferentiated cells.** SH-SY5Y cells were cross-linked using formaldehyde. Ribosomal cross-links observed in undifferentiated cells were visualised on the high-resolution structure of the ribosome (PDB ID: 4UG0[57]). 43 cross-links have a Cα distance < 30 Å (blue), one cross-link has a Cα distance >30 Å (red). Figure generated using Supplementary Data 4.

Although neuronal cell cultures are often limited in the available sample amount, we successfully analysed the proteome of three different cell-types and identified more than 3500 proteins in 18 replicates (i.e. six biological replicates per cell culture condition). A total of almost 6000 identified proteins confirms a good coverage of the cellular proteome. Even though in-cell cross-linking experiments are often limited to the high-abundant proteins expressed in a cell-type[73], they are advantageous over other approaches: First, the interactions are analysed in a native cell environment and artificial labels or genetic modifications that might affect interactions are not required. Similarly, cell disruption and the addition of protease inhibitors are performed after introducing the linkages; stability of the protein assemblies is, therefore, ensured during sample preparation. As a result, we successfully visualised structural differences between undifferentiated and differentiated cells. Our stringent thresholds applied during data analysis assure a low number of false positives as confirmed by the presence of only one decoy hit. Structural validation further confirmed the applicability of our protocols which will be of interest for future studies using cell cultures at low scale. Note that, even though protein interactions were identified with high-confidence when following our approach, the absence of cross-links does not confirm that these interactions are not formed in the protein assemblies.

The proteome and KEGG pathway analysis of undifferentiated SH-SY5Y cells uncovered typical cancer characteristics demonstrated by constant proliferation and cancer progression. In agreement, DNA replication was previously found to be inhibited after 24 hours of treatment with RA[74]. The explicit functions of upregulated, cancer-related proteins in neuroblastoma are subject to future studies. The applicability of undifferentiated SH-SY5Y cells for studying neuronal function and dysfunction, therefore, remains to be discussed.

RA- and RA/PMA-differentiated SH-SY5Y cells showed changes in the morphology and in the expression of proteins required for structural and functional changes of the cells.

However, only minor differences in protein abundances and expression profiles were identified between the two cell-types. Similar to a previous study comparing the proteomes of GABAergic and glutamergic synaptic vesicles and synaptic docking complexes[75,76] differences in their protein expression profiles are mostly expected for their neurotransmitter transporters (see below). Overall, our results are in agreement with a previous study relatively quantifying proteome remodelling during differentiation with RA[13]. Zhang et al. examined temporal expression patterns of specific proteins. For instance, cytochrome P450 26B1 (CYP26B1), neural cell adhesion molecule 2 (NCAM2) and extracellular leucine-rich repeat and fibronectin type III (ELFN1) were upregulated upon RA-differentiation; accordingly, we identified CYP26B1 and NCAM2 only in differentiated cells. Likewise, rabphilin-3A (RPH3A) was found to be upregulated in undifferentiated cells; we identified this protein only in undifferentiated cells. Several neuron-specific proteins such as PDZ and LIM domain protein 5 or integrin alpha1 were identified in both studies. Importantly, Zhang et al. could not identify a complex time-dependent mechanism of RA-differentiation[13]. These findings correlate well with our observation that RA-differentiated SH-SY5Y cells do not develop a defined neuronal sub-type (see below). A similar study, employing a combination of RA and BDNF for differentiation of SH-SY5Y cells, was presented by Murillo et al.[3]. In this study, several proteins including, for instance, cellular retinoic acid-binding protein 2, integrin alpha-1, integrin beta-1 or filamin-B were found to be upregulated upon differentiation; these proteins were also upregulated when following our differentiation protocols. However, the highly upregulated MARCKS-related protein using RA/BDNF for differentiation was slightly downregulated in our experiments suggesting some differences in the differentiation process when BDNF is employed.

In our proteomics experiments, we did not apply specific protocols for integral membrane proteins and we, therefore, likely missed the various transporters in these analyses. For

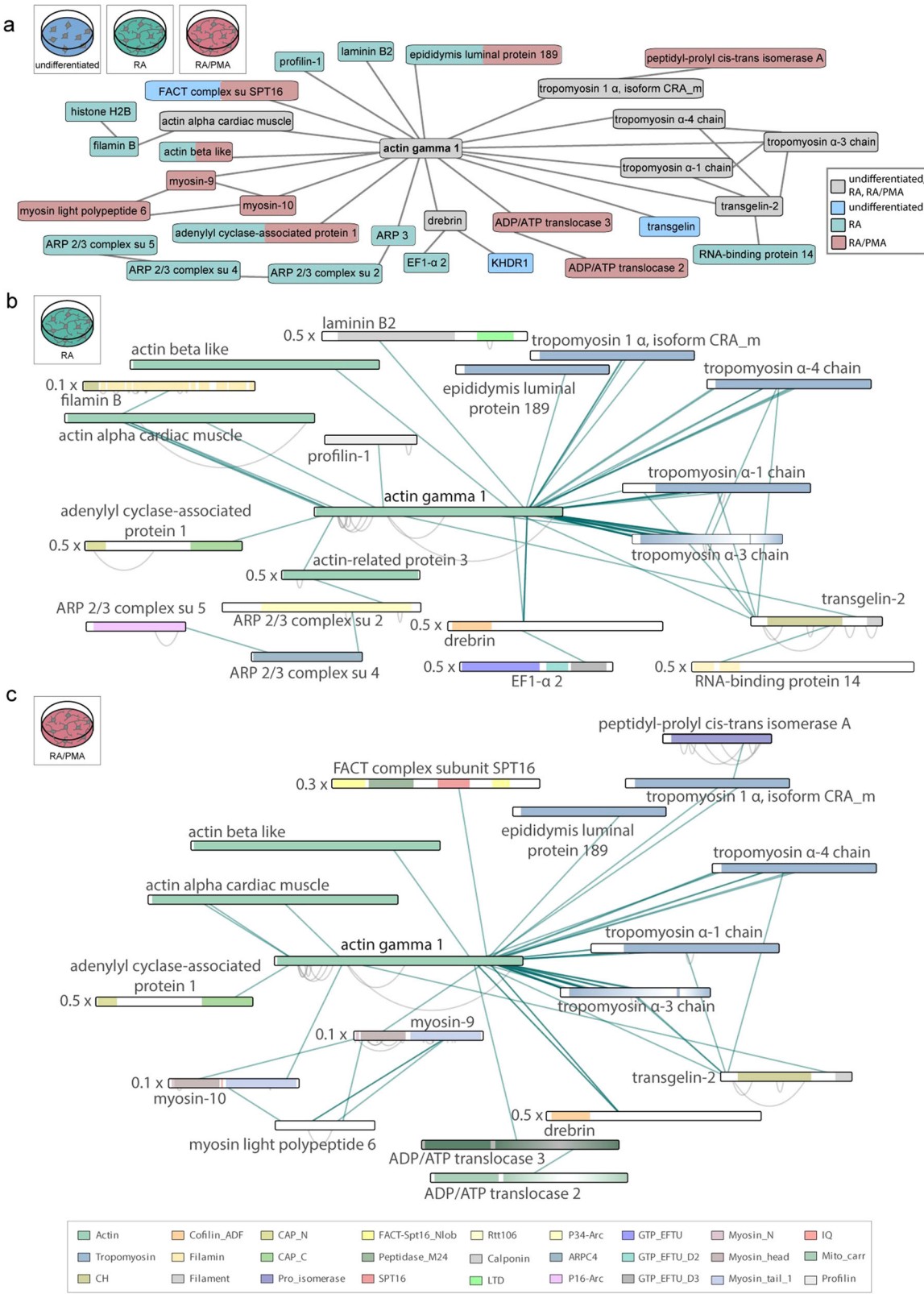

characterisation of neuronal sub-types, we, therefore, performed qPCR measurements. As expected from our proteome analysis, undifferentiated cells did not show a specific neuronal sub-type. For differentiated cells, however, we observed overexpression of specific marker genes. Previous studies suggested that RA-differentiation results in formation of adrenergic, dopaminergic and predominantly cholinergic neuronal subtypes[1,2]. Here, we

clearly observed development of cholinergic and acetylcholinergic sub-types; expression of dopaminergic markers did not show a clear tendency. Following RA/PMA-differentiation, acetylcholinergic neuronal subtypes were obtained, while dopaminergic markers were again not clearly upregulated. We conclude that RA-differentiation results in a mixture of neuronal sub-types. RA/PMA-differentiated cells, on the other hand, show a clear

**Fig. 7 Protein interaction networks of actin gamma 1.** Proteins are represented as bars. N-termini and C-termini correspond to left and right side of the bars. Note that some proteins are scaled (indicated by a scaling factor). Inter-molecular protein interactions are indicated by lines. **a** Interaction network of actin-gamma 1 cross-linked proteins. Interactions observed in the three cell culture conditions are indicated (see legend for details). Note that residue information is not included. **b** Detailed actin gamma 1-network observed in RA-differentiated cells. Protein domains are indicated (see legend for details). **c** Detailed actin gamma 1-network observed in RA/PMA-differentiated cells. Protein domains are indicated (see legend for details). CH calponin homology, ADF actin depolymerisation factor, CAP_N adenylate cyclase-associated N terminal, CAP_C adenylate cyclase-associated C terminal, Pro_isomerase peptidyl-prolyl cis-trans isomerases, FACT-Spt16_Nlob FACT complex subunit SPT16 N-terminal lobe domain, Peptidase_M24 metallopeptidase family M24, SPT16 FACT complex subunit (SPT16/CDC68), Rtt106 Histone chaperone Rttp106-like, LTD lamin tail domain, P34-Arc Arp2/3 complex 34 kD subunit p34-Arc, ARPC4 ARP2/3 complex 20 kDa subunit (ARPC4), P16-Arc ARP2/3 complex 16 kDa subunit (p16-Arc), GTP_EFTU GTP-binding elongation factor family EF-Tu/EF-1A subfamily, GTP_EFTU_D2 elongation factor Tu domain 2, GTP_EFTU_D3 elongation factor Tu C-terminal domain, Myosin_N myosin N-terminal SH3-like domain, IQ Q calmodulin-binding motif, Mito_carr mitochondrial carrier protein. Figure generated using Supplementary Data 4.

tendency for expression of the acetylcholine marker. Depending on the research question to be answered, the development of different neuronal sub-types should be considered when choosing a differentiation protocol.

In addition to protein expression, post-transcriptional modifications might also play a role in the differentiation of SH-SY5Y cells bringing about differences between neuronal sub-types which have not been explored in our analyses. Phosphorylation is a prevalent modification and was analysed in RA/BDNF differentiated SH-SY5Y cells[3]. Additional post-translational modifications such as acetylation[77] or glycosylation of synaptic proteins[78] are important for specific neuronal function and should be addressed in future studies.

The only differences observed in the proteomes of RA- and RA/PMA-differentiated SH-SY5Y cells are related with the lysosome. Both, the subcellular localisation and KEGG pathways analysis, revealed a significantly higher expression of lysosomal proteins in RA/PMA-differentiated cells. The main function of the lysosome is the degradation of cellular components suggesting higher recycling activities when inducing differentiation additively with a second chemical components. A rather undefined neuronal sub-types for RA-differentiated SH-SY5Y cells was previously described[1,2] and confirmed by our own results (see above) suggesting that continued differentiation with PMA includes the turnover of specific proteins such as neurotransmitter transporters. A higher lysosomal activity is therefore required during differentiation of these neuron-like cells.

Note that we did not observe overexpression of mature neuronal markers such as microtubule-associated protein 2 or β-III tubulin. Upregulation of nestin and other proteins required for neuronal differentiation suggest that the cells are in the state of early differentiation. Downregulation of doublecortin further suggests that the cells are possibly in an intermediate state between early and mature neurons. Applying longer differentiation protocols might produce cells with mature neuronal characteristics and might further support differentiation into specific neuronal sub-types (see above). We conclude that SH-SY5Y cells obtained after 5 days of RA or 6 days of RA/PMA treatment represent model systems for early differentiating states. These early differentiating cells showed overexpression of proteins associated with RA treatment, neuronal differentiation, establishment of synapses and antioxidant defence as well as proteins regulating the actin cytoskeleton.

Using formaldehyde for in-cell cross-linking, we obtained an interaction network of actin gamma 1 and additional proteins. Importantly, interactions with actin-regulating proteins were predominantly observed upon differentiation. The absence of these interactions in undifferentiated cells suggests reorganisation of the actin cytoskeleton during differentiation. Interactions with actin dynamic regulating proteins such as the ARP2/3 complex, myosin variants or the cyclase-associated protein 1 observed in

differentiated cell types suggest a specific function during early neuronal development associated with actin reorganisation.

## Methods

**Cell culture**. The human cell line SH-SY5Y was obtained from Deutsche Sammlung von Mikroorganismen und Zellkulturen (DSMZ). The cells were cultured in minimum essential medium eagle (Sigma-Aldrich) supplemented with 15% (v/v) fetal bovine serum (ATCC), 1% (v/v) penicillin/streptomycin (Life Technologies) and 2 mM glutamine in an atmosphere of 5% $CO_2$ at 37 °C. The medium was changed every two to three days by passaging the cells when a confluence of ~80% was reached.

**Differentiation of SH-SY5Y cells**. For differentiation, cells with a passage number <8 were employed. Two differentiation media were used: First, minimum essential medium eagle supplemented with 2.5% (v/v) fetal bovine serum, 1% (v/v) penicillin/streptomycin, 2 mM glutamine and 10 μM RA (differentiation medium 1) and, second, minimum essential medium eagle supplemented with 2.5% (v/v) fetal bovine serum, 1% (v/v) penicillin/streptomycin, 2 mM glutamine, 10 μM RA and 80 μM PMA (differentiation medium 2). For proteome analysis, 100,000 cells were plated in six-well plates. For cross-linking analysis, 800,000 cells were plated in T75 flasks. For RA differentiation, the medium was exchanged for differentiation medium 1 after 1, 3 and 5 days followed by cell harvest or cross-linking on day 6. For RA/PMA-differentiation, the medium was exchanged for differentiation medium 1 on days 1 and 3 and for differentiation medium 2 on days 4 and 6. Cells were cross-linked and/or harvested on day 7. An inverted Primovert light microscope (Zeiss) at ×20 magnification equipped with an 18 MP microscope digital camera (Swift Optical Instruments, Inc.) was used to monitor cell growth and differentiation. Undifferentiated cells were detached from the plate using trypsin-ethylenediaminetetraacetic acid (0.5%, Gibco) followed by repeated washing with phosphatebuffered saline (PBS). The cell pellet was stored at −80 °C.

**Cross-linking of proteins with formaldehyde**. For cross-linking, cells were washed three times with PBS. Subsequently, 3 ml of 4.5% (v/v) formaldehyde in PBS were added to the cells followed by incubation for 15 min at 37 °C. The cells were washed three times with ice-cold PBS, scraped off the T75 flask into PBS and pelleted by centrifugation at $800 \times g$ for 2 min at 4 °C.

**Sample preparation for LC-MS/MS**. Cell-lysis of non-cross-linked cells was performed as previously described[79]. Briefly, 4 vol. of trifluoroacetic acid (TFA) were added to the cell pellet followed by incubation for approx. 2 min at room temperature until cells were completely lysed. Subsequently, 10 vol. (corresponding to the amount of TFA used for cell lysis) of 2 M tris(hydroxymethyl)aminomethan were added for neutralisation followed by addition of 1.1 vol. (according to the amount of TFA used for cell lysis) of 29 mM tris(2-carboxyethyl)phosphine (TCEP) and 37 mM 2-chloroacetamide (CAA). After addition of TCEP and CAA, the cell lysate was incubated for 5 min at 95 °C. The cell lysate was diluted 1:5 with water and trypsin was added at a protein:enzyme ratio of 50:1. The protein amount was estimated based on the starting material after cell lysis. Tryptic hydrolysis was performed overnight at 37 °C and 600 rpm in a thermomixer. Subsequently, the peptides were desalted using Pierce Peptide Desalting Spin Columns (Thermo Fisher) (see below).

For cell-lysis of cross-linked samples, the protocol was slightly adapted to avoid high temperatures: 10 vol. of 2 M Tris were added on ice; reduction with TCEP and alkylation with CAA was performed at 37 °C for 15 min and 300 rpm (see above for further details). The proteins were then digested on Sera-Mag SpeedBeads (GE Healthcare) according to a previously published protocol[80]. Briefly, the beads where washed three times with water and the cell lysate was added to the beads followed by addition of 1 vol. 100% (v/v) ethanol. The mixture was incubated for 5 min at 24 °C and 1000 rpm. The supernatant was discarded by fixation of the beads using a magnet. The beads and bound protein were washed three times with

80% (v/v) ethanol. Subsequently, trypsin (Promega) in 25 mM ammonium bicarbonate, pH 8.5 was added at an enzyme:protein ratio of 1:50. The sample was sonicated for 30 s in a water bath to disaggregate the beads followed by incubation in a thermomixer at 37 °C and 1000 rpm overnight. Generated peptides were recovered by centrifugation at $20,000 \times g$ for 1 min at room temperature.

**Peptide desalting.** The peptides were desalted using Pierce™ Peptide Desalting Spin columns (Thermo Scientific) according to the manufacturer's protocol. Briefly, the storage buffer was removed by centrifugation at $5000 \times g$ for 1 min. The spin column was washed twice by adding 300 µl acetonitrile (ACN) followed by centrifugation at $5000 \times g$ for 1 min. The column was again washed twice with 0.1% (v/v) TFA as described. The peptides were then sequentially loaded onto the spin column in 300 µl volumina followed by centrifugation at $3000 \times g$ for 1 min after each loading step. After washing the column three times with 0.1% (v/v) TFA as described above, desalted peptides were eluted with 300 µl 50% (v/v) ACN and 0.1% (v/v) TFA by centrifugation at $3000 \times g$ for 1 min. Elution was repeated once and the peptides were dried in a vacuum centrifuge.

**Enrichment of cross-linked peptide pairs.** Peptides of cross-linked proteins were dissolved in 30% (v/v) ACN, 0.1% (v/v) TFA. Cross-linked peptide pairs were enriched by peptide size exclusion chromatography using an ÄKTA pure chromatography system (GE Healthcare) equipped with a Superdex™ peptide 3.2/300 column (GE Healthcare). Peptides were separated isocratically with a flow rate of 50 µl/min using 30% (v/v) ACN, 0.1% (v/v) TFA as mobile phase. Elution of peptides was monitored at 280 nm. Fractions of 50 µl containing cross-linked peptide pairs (early fractions) as well as linear peptides (late fractions) were collected. The peptides were dried in a vacuum centrifuge.

**LC-MS/MS.** Tryptic peptides were dissolved in 2% (v/v) ACN/0.1% (v/v) FA and analysed by nano-flow reversed-phase liquid chromatography on a Dionex-UltiMate 3000 RSLCnano System (Thermo Scientific; mobile phase A, 0.1% (v/v) formic acid (FA); mobile phase B, 80% (v/v) ACN/0.1% (v/v) FA) coupled with a Q Exactive Plus Hybrid Quadrupole-Orbitrap mass spectrometer (Thermo Scientific). For desalting, peptides were loaded onto a trap column (µ-Precolumn C18 Acclaim™ PepMap™ 100, C18, 300 µm I.D., particle size 5 µm; Thermo Scientific) with a flow rate of 10 µl/min. The peptides were then separated with a flow rate of 300 nL/min on an analytical C18 capillary column (50 cm, HPLC column Acclaim™ PepMap™ 100, C18, 75 µm I.D., particle size 3 µm; Thermo Scientific). For proteome analysis, a gradient of 4–90% (v/v) mobile phase B over 152 min was applied. For analysis of cross-linked peptides, a gradient over 92 min was applied and adjusted step-wise for early, middle and late fractions of the peptide size exclusion chromatography. Peptides were directly eluted into the mass spectrometer.

Typical mass spectrometric conditions were: spray voltage, 2.8 kV; capillary temperature, 275 °C; data-dependent and positive ion mode. For proteome analysis, survey full scans were acquired in the orbitrap with a resolution of 70,000, an automatic gain control (AGC) target of 3e6, a maximum injection time of 80 ms and a scan range from 350 to 1600 m/z. Fragmentation of the 20 most intense ions with charge states of 2+ to 7+ was performed in the HCD cell employing a normalised collisional energy of 30%. MS/MS spectra were acquired with a resolution of 17,500, an AGC target of 1e5 and a maximum injection time of 150 ms. The fixed first mass was set to 105 m/z. Previously selected ions were dynamically excluded for 30 s.

For analysis of cross-linked proteins, survey full scans were acquired in the orbitrap with a resolution of 70,000, an AGC target of 3e6, a maximum injection time of 100 ms and a scan range from 350 to 1600 m/z. Fragmentation of the 20 most intense ions with charge states of 4+ to 7+ was performed in the HCD cell employing a stepped collisional energy of 30%. MS/MS spectra were acquired with a resolution of 35,000, an AGC target of 1e5 and a maximum injection time of 200 ms. The fixed first mass was set to 105 m/z. Previously selected ions were dynamically excluded for 30 s. The lock mass option (lock mass m/z 445.120025) was enabled in all measurements[81].

**Proteomic data analysis.** Raw data of 6 biological replicates of each culture condition were searched against the human uniprot database (Uniprot, Proteome ID: UP000005640, 73947 entries, version date: 1st April 2019) using MaxQuant (version 1.6.17.0)[19]. Standard parameters were used: fixed modification, carbamidomethyl (cysteine); variable modifications, oxidation (methionine) and acetylation (protein N-terminus); max missed cleavage sites, 2; min peptide length, 7; max peptide mass 6,000 Da; peptide FDR, 0.01; protein FDR, 0.01; enzyme, trypsin/P (cleavage C-terminal of lysine or arginine also when the C-terminal amino acid is proline). The iBAQ, MaxQuant LFQ and 'match between runs' options were enabled.

For calculation of the subcellular localisation of proteins, iBAQ values were used. For this, the subcellular localisation of proteins with known main localisation was first assigned using the localisation of Beltran et al as reference[18]. The sum of iBAQ values of proteins of the same localisation was then calculated for each experiment. Finally, the mean and standard error for each localisation and culture condition was calculated. Significant differences between the culture conditions for

each localisation were determined by performing a t-test. Adjusted p-values were obtained using the Bonferroni correction method[82].

Quantification of global protein abundances was performed in RStudio (version 1.2.1335) using R (version 4.0.2) with artMS (version 1.9.1) (http://artms.org) and MSstats (version 3.22.0)[83]. The following parameters were used: relative quantification method, global protein abundance; normalisation method, equalise medians; cutoffCensored, minFeature; annotation species, human. For relative global protein abundance quantification of the different cell culture conditions, proteins with a log2(fold-change) < −0.8 or >0.8 and an adjusted p-value < 0.05 according to Benjamini and Hochberg were considered as significantly up- or downregulated[84].

**Cross-linking data analysis.** For cross-linking analysis, raw data files were converted to mascot generic file format (mgf) using Thermo Proteome Discoverer (version 2.4.1.15). For each cell culture condition, a specific database was generated using the 800 most abundant proteins (according to the mean iBAQ of 6 replicates). Cross-linking analysis was performed using 'Formaldehyde Cross-link Analyser'[14] with the following parameters: MS1 tolerance, 6 ppm; MS2 tolerance, 8 ppm; number of missed cleavage sites, 4; 'use decoy sequence' was enabled. Cross-linked peptides have a characteristic mass shift of 12 or 24 Da[14,85]. The results table was filtered for high and intermediate confident cross-links. Cross-links with a ratio of (# observed fragments / # amino acids of the cross-linked peptide pair) >1.5 as well as a total number of fragments for each peptide >18 were considered high-confident cross-links[14]. Cross-links with a total number of fragments for each peptide >15 were considered intermediate confident cross-links[14]. The overlap of high and intermediate confident cross-links identified in the three replicates was analysed using the web-based venn-tool (http://bioinformatics.psb. ugent.be/webtools/Venn/). The following scores were assigned: high confident cross-link in three replicates = 1; high confident cross-link in two replicates = 2; high confident cross-link in one replicate and at least one intermediate confident cross-link in another replicate = 3; intermediate confident cross-link in three replicates = 4; intermediate confident cross-link in two replicates = 5, high confident cross-link in one replicate = 6, intermediate confident cross-link in one replicate = 7. Visualisation of cross-links with a score ≤4 was performed in Cytoscape (version 3.8.2)[86] using the Cytoscape app XlinkCyNET (version 1.2.5)[87]. Analysis of cross-links in high-resolution structures was performed using UCSF Chimera (version 1.15)[88] and the software tool Xlink Analyzer (version 1.1.4)[89].

**Isolation of RNA and cDNA synthesis.** The RNA of SH-SY5Y cells was purified using the GeneMATRIX Universal RNA Purification Kit (Roboklon) according to the manufacturer's protocol. Briefly, pelleted SH-SY5Y cells were lysed with 400 µl RL buffer. The cell lysate was transferred to the homogenisation spin column and centrifuged at $11,000 \times g$ for 2 min. 250 µl 100% (v/v) ethanol were added to the flow-through and the flow-through was transferred to the RNA binding spin column and centrifuged for 1 min at $11,000 \times g$. The flow-through was discarded. 400 µl Wash DN1 buffer were added followed by centrifugation for 1 min at $11,000 \times g$. Subsequently, 600 µl Wash RBW buffer were added, and the sample was centrifuge for 1 min at $11,000 \times g$. For the next washing step, 300 µl Wash RBW buffer were added, and the sample was centrifuge for 2 min at $11.000 \times g$. The RNA was eluted from the membrane in 40 µl RNase-free water and centrifugation for 1 min at $11,000 \times g$. The concentration and quality of the RNA purification was determined at 260 nm and ratio of 260/280 nm and 260/230 nm. For cDNA synthesis, the qScript cDNA SuperMix (Quantabio) was added to 1 µg RNA and incubated for 5 min at 25 °C, for 30 min at 42 °C and for 5 min at 85 °C. cDNA was stored at –20 °C.

**Quantitative polymerase chain reaction.** The Quant Studio 3™ Real Time System and QuantStudio™ Design & Analysis Software v1.5.2 (Thermo Scientific) were used for qPCR analysis. Primer pairs *HPRT1*, *TGM2*, *SYN1*, *RET*, *DBH*, *DRD2*, *CHT* and *ACHE* were selected (see Supplementary Data 3). Six biological replicates were performed for each cell culture condition of SH-SY5Y cells (undifferentiated, RA- and RA/PMA-differentiated cells). Each biological replicate was analysed in technical duplicates. No template controls were used for each primer set. For qPCR, 3 µl cDNA (corresponding to 38 ng RNA) and 17 µl MasterMix (Power-Track™ SYBR Green Master Mix, Thermo Scientific) were mixed. Optimised conditions were applied for each primer pair (Supplementary Data 3). Each reaction was subjected to melting temperature analysis to confirm presence of specific singly amplified products (Supplementary Fig. 8). The primer efficiency was calculated using a dilution series of cDNA as template (Supplementary Data 3). Specific gene amplification was normalised to *HPRT1* and quantification was performed using the $2^{-\Delta\Delta Ct}$ method[53]. To calculate the fold-change the geometric mean of undifferentiated cells was used. A two-tailed *t*-test between two conditions was used for calculation of *p*-values.

**Statistics and reproducibility.** All proteomic analyses were performed in six independent replicates for each cell culture condition. For qPCR experiments six biological replicates were analysed in technical duplicates. For cross-linking experiments three independent replicates were used. The sum of iBAQ values of all proteins linked with the same localisation are presented as mean with standard

error. Individual data points for each localisation and condition are given; adjusted *p*-values <0.01 (Bonferroni correction method[82]) were considered significant. For relative global protein abundance quantification of the different cell culture conditions a two-tailed t-test was performed. Adjusted *p*-values were calculated according to Benjamini and Hochberg[84]. Proteins with a log2(fold-change) <−0.8 or >0.8 and adjusted *p*-values <0.05 were considered to be significantly up- or downregulated. For statistical data analysis of qPCR experiments, the gene amplification was normalised to HPRT1 and quantification was performed using the $2^{-\Delta\Delta Ct}$ method[53]. To calculate the fold-change, the geometric mean of undifferentiated cells was used. qPCR data are presented as mean with standard error. Individual data points are given and *p*-values <0.05 were considered significant.

**Reporting summary**. Further information on research design is available in the Nature Research Reporting Summary linked to this article.

## Data availability

All MS raw files and the corresponding results files including databases were deposited to the ProteomeXchange Consortium (www.proteomexchange.org) via the PRIDE[90] partner repository with the dataset identifier PXD031054. The source data underlying the graphs and charts are provided in Supplementary Data 1 (Fig. 2), Supplementary Data 2 (Fig. 3), Supplementary Data 3 (Fig. 4) and Supplementary Data 4 (Figs. 5–7). All other data are available from the corresponding author upon request.

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

## Acknowledgements
We thank Leonie Jaster and Lara Meret Peters for preliminary cell culture experiments. This work was funded by the Federal Ministry for Education and Research (BMBF, ZIK programme, 03Z22HN22), the European Regional Development Funds (EFRE, ZS/2016/04/78115) and the MLU Halle-Wittenberg. ATN and ANS acknowledge funding from the German research Foundation (DFG; Research Training Group 2155 'ProMoAge', project number 270489335).

## Author contributions
M.B. and C.S. designed the research, M.B. performed all proteomics and cross-linking experiments and data analysis, M.B. and A.T.N. performed qPCR experiments, A.N.S. supervised qPCR experiments, C.S. supervised proteomics and cross-linking experiments and guided the research, M.B. and C.S. wrote the manuscript with contribution from all authors.

## Funding

## Competing interests
The authors declare no competing interests.
