## [Peer Review File · Communications Biology]

Reviewers' comments:

Reviewer #1 (Remarks to the Author):

Review of "Quantitative proteomics and in-cell cross-linking reveal cellular reorganization during early neuronal differentiation of SH-SY5Y cells induced by retinoic acid or phorbol-12-myristat-13-acetate"

(COMMSBIO-21-1642-T) by Marie Barth and Carla Schmidt

In this article, the authors investigated the differentiation of SH-SY5Y cells with retinoic acid (RA) and phorbol-12-myristat-13-acetate (PMA). They compared the proteome-wide changes of two differentiation protocols using RA and RA/PMA and highlighted proteins involved in the early differentiation of neurons and marker proteins.

Overall, this article is well done. The article is concisely and clearly written. The experiments selected address well the questions proposed, and the results provide novel insight into the differentiation of SHSY5Y with RA and RA/PMA. However, some minor revisions are needed before I can recommend his manuscript for publication in Communications Biology.

-In order to orientate better the reader, I suggest an additional figure, before Figure 1, which outlines the workflow. I suggest basically an expanded version of the Graphical Abstract with a descriptive caption.

-Supplemental Figure 2. The upset plot is a nice touch, but it would be also interesting to tally up the number of proteins common to 16 samples, 15 samples, 14 samples, etc. in the form of a bar graph. This graph could seamlessly be inserted as an inset in the upper left of this figure.

-In the methods section page 21, line 506 states "a stepped collision energy of 30%." What are the steps (i.e., the other collision energy aside from 30%)?

-More of a matter of aesthetics, but in figure 1, the peroxisome panel should have its y-axis in scientific notation much like how it is done on the Golgi panel.

-Few typos throughout including "extend" on pg 6 line 142, which should be "extent." "acetonitril" on page 20, line 469.

-The most significant revision deals with the comparison of the author's work with that of similar studies. The authors should provide a bit more comparison between their work and that of Murillo et al (ref 3) and Zhang et al (ref 13) with respect to RA differentiation and highlighted proteins involved in the early differentiation of neurons, as both these work feature the proteomic analysis of differentiation in SH-SY5Y cells.

Reviewer #2 (Remarks to the Author):

The provided manuscript by Barth and Schmidt provides a descriptive proteomic analysis of SH-SY5Y cells, a cell model for neuronal differentiation. Lysates from cells treated with RA, RA+PMA and controls were compared with each other on protein abundance and protein complex level, the latter enabled by using formaldehyde crosslinking followed by mass spectrometric analysis.

As SH-SY5Y cells are widely used as model for neuronal differentiation, a detailed characterization of this model is an important task. The provided manuscript therefore provides valuable data for a more complete characterization of this cell model. The manuscript is good to read and easy to follow and my impression is that the overall technical quality is very good. From my point of view, the strength of the manuscript is the provided data on protein complexes even is most of the identified complexes include higher abundant proteins. Some work on relative proteome abundance changes in SH-SY5Y cells has been published before at least for RA differentiation.

From my point of view, the manuscript would profit from

1. A clearer focus on the protein complex/crosslink data
2. A more stringent experimental setup (controls, timepoints)
3. A functional/physiological characterization of differences between cells differentiated with different protocols at least if the claim is made the different sub-types of differentiated cells are compared.
4. A validation of at least some of the highlighted protein complexes with an alternative method to prove if there is really a change in complex composition / protein-protein interaction of differentiated vs. undifferentiated cells.
5. A comparison of the protein abundance data to comparable studies e.g. by Zhang et al. to make the added value of this study clearer.

Major remarks

1. Line 58: "...a comparative proteome analysis of the various sub-types is still missing". There are several publications (e.g. the cited Zhang et al. and others which were not cited) which show a comparative proteome analysis of RA differentiated and undifferentiated SH-SY5Y cells. Some of those provide even a greater depth, find more differences and/or include more timepoints than 5/6-day differentiation. If the claim is that additionally different sub-types are compared, my feeling is that it would be important to show, that really different subtypes have been analyzed e.g. by functional/physiological assays. It is written that "Differences in the phenotypes of the cells obtained using the two differentiation protocols have not been observed". Which assays have been used to characterize the phenotypes that no differences were obvious? Shouldn't RA/PMA lead to a differentiation into dopaminergic-like neurons as written in the introduction?
2. The experimental setup is not totally clear to me (and maybe not straight forward). It is written, that undifferentiated cells were grown under standard conditions. Does this include that there were grown in 15% FBS? If this would be the case, it is not clear which effect the change of the serum concentration has on the cells and if the found differences at least in part might be due to the adaption to lower-serum conditions. This should at least be controlled e.g. by cells cultivated in exactly the same medium (2.5% FCS, solvent control?) with and without t RA / RA+PMA treatment. Furthermore, the differentiation protocols differ in length (6 days RA, 7 days RA+PMA). How can the authors be sure that the effect on the proteome really results from additional PMA treatment and not from the longer incubation time? What about the control? Have the control cells here been cultivated for 6 days or 7 days without passaging?
3. Line 68: "...and identify changes in the proteins' subcellular localization." As it is written here, I would expect a localization change of proteins within the cell. No experiment has been made to support that. Furthermore, if I understand the approach right, a very prominent abundance change of only one protein theoretically might be sufficient to give in sum a significant change. Would it be possible to either comment on this or to show some data e.g. the proportion of intensities of single proteins to address this?
4. The discussion is partly redundant and partly a repetition of the results. I'd suggest to remove the redundant parts and include instead e.g. a comparison to already published datasets e.g. Zhang et al. It would also be interesting which protein complexes have been described before and which might be novel.
5. Is there any reason that much more complexes were identified in differentiated cells or might this be an artifact e.g. through the higher amount of serum which has not been completely washed away before crosslinking? This might be checked by including also potential serum contaminants into the search. Has this been done for the crosslink data? In the protein abundance data, the last three replicates of both differentiated cells include about the double amount of potential serum contaminants. Is there a reason for this?

Minor remarks

1. line 30: "... reveal the presence of early differentiating neurons". I'd be cautious here; my feeling is that it quite difficult or impossible to show with the used methods that there are really neurons in the dish. If this claim is made, it should to be proven at least by functional/physiological assays.
2. Proteome analysis: Did I get it right, that only proteins were considered showing no missing

value in all replicates of the respective comparisons? Were proteins identifications on the basis of only one peptide included?

3. Line 128: What is meant with "significance threshold of 5%?"

4. Legend Figure2: Should be "-log10"...

5. Line 232: "We conclude that undifferentiated SH-SY5Y cells represent neuroblastoma cells...". That's not really a big surprise as this is already clear at least from the introduction...

6. Line 449: a protein:enzyme ratio 50:1 is mentioned. How has the protein concentration/amount been determined?

Reviewers' comments:

Reviewer #1 (Remarks to the Author):

Review of “Quantitative proteomics and in-cell cross-linking reveal cellular reorganization during early neuronal differentiation of SH-SY5Y cells induced by retinoic acid or phorbol-12-myristat-13-acetate”

(COMMSBIO-21-1642-T) by Marie Barth and Carla Schmidt

In this article, the authors investigated the differentiation of SH-SY5Y cells with retinoic acid (RA) and phorbol-12-myristat-13-acetate (PMA). They compared the proteome-wide changes of two differentiation protocols using RA and RA/PMA and highlighted proteins involved in the early differentiation of neurons and marker proteins.

Overall, this article is well done. The article is concisely and clearly written. The experiments selected address well the questions proposed, and the results provide novel insight into the differentiation of SHSY5Y with RA and RA/PMA. However, some minor revisions are needed before I can recommend his manuscript for publication in Communications Biology.

-In order to orientate better the reader, I suggest an additional figure, before Figure 1, which outlines the workflow. I suggest basically an expanded version of the Graphical Abstract with a descriptive caption.

We thank the reviewers for this suggestion. We included a workflow in our revised manuscript (Figure 1). This workflow provides an overview on all performed experiments (cell differentiation, proteomics and cross-linking).

-Supplemental Figure 2. The upset plot is a nice touch, but it would be also interesting to tally up the number of proteins common to 16 samples, 15 samples, 14 samples, etc. in the form of a bar graph. This graph could seamlessly be inserted as an inset in the upper left of this figure.

We agree. We included a bar digramme providing the number of proteins identified in 18, 17, 16, etc. samples.

-In the methods section page 21, line 506 states “a stepped collision energy of 30%.” What are the steps (i.e., the other collision energy aside from 30%)?

We carefully reviewed the parameters and found that a normalized collisional energy of 30 % was used. We apologize for this confusion.

-More of a matter of aesthetics, but in figure 1, the peroxisome panel should have it's y-axis in scientific notation much like how it is done on the Golgi panel.

We agree. We have changed this figure accordingly (Figure 2 of the revised manuscript).

-Few typos throughout including “extend” on pg 6 line 142, which should be “extent.” “acetonitril” on page 20, line 469.

We apologize for these mistakes. We have corrected the typos and also carefully checked the entire manuscript.

-The most significant revision deals with the comparison of the author's work with that of similar studies. The authors should provide a bit more comparison between their work and that of Murillo et al (ref 3) and Zhang et al (ref 13) with respect to RA differentiation and highlighted proteins involved in the early differentiation of neurons, as both these work feature the proteomic analysis of differentiation in SH-SY5Y cells.

We included a paragraph comparing our results with the studies of Murillo et al. and Zhang et al. in the discussion section of our revised manuscript.

Reviewer #2 (Remarks to the Author):

The provided manuscript by Barth and Schmidt provides a descriptive proteomic analysis of SH-SY5Y cells, a cell model for neuronal differentiation. Lysates from cells treated with RA, RA+PMA and controls were compared with each other on protein abundance and protein complex level, the latter enabled by using formaldehyde crosslinking followed by mass spectrometric analysis.

As SH-SY5Y cells are widely used as model for neuronal differentiation, a detailed characterization of this model is an important task. The provided manuscript therefore provides valuable data for a more complete characterization of this cell model. The manuscript is good to read and easy to follow and my impression is that the overall technical quality is very good. From my point of view, the strength of the manuscript is the provided data on protein complexes even is most of the identified complexes include higher abundant proteins. Some work on relative proteome abundance changes in SH-SY5Y cells has been published before at least for RA differentiation.

From my point of view, the manuscript would profit from

1. A clearer focus on the protein complex/crosslink data
2. A more stringent experimental setup (controls, timepoints)
3. A functional/physiological characterization of differences between cells differentiated with different protocols at least if the claim is made the different sub-types of differentiated cells are compared.
4. A validation of at least some of the highlighted protein complexes with an alternative method to prove if there is really a change in complex composition / protein-protein interaction of differentiated vs. undifferentiated cells.
5. A comparison of the protein abundance data to comparable studies e.g. by Zhang et al. to make the added value of this study clearer.

We thanks the reviewer for these suggestions. We have addressed these issues in detail in the following points.

Major remarks

1. Line 58: "...a comparative proteome analysis of the various sub-types is still missing". There are several publications (e.g. the cited Zhang et al. and others which were not cited) which show a comparative proteome analysis of RA differentiated and undifferentiated SH-SY5Y cells. Some of those provide even a greater depth, find more differences and/or include more timepoints than 5/6-day differentiation. If the claim is that additionally different sub-types are compared, my feeling is that it would be imported to show, that really different subtypes have been analyzed e.g. by

functional/physiological assays. It is written that “Differences in the phenotypes of the cells obtained using the two differentiation protocols have not been observed”. Which assays have been used to characterize the phenotypes that no differences were obvious? Shouldn't RA/PMA lead to a differentiation into dopaminergic-like neurons as written in the introduction?

We agree that there are several proteome studies on RA differentiation (e.g. Zhang et. al.). While some of the results presented in these studies are similar to our study, our comparison delivered additional insights, mainly with respect to structural rearrangements in the cells during neuronal differentiation. Note that we included an additional, commonly applied differentiation protocol employing RA+PMA in our study. We modified the statement in Line 58 accordingly. It now reads: “Although many studies commonly employ undifferentiated or differentiated SH-SY5Y cells as neuronal model system, the comparative proteome analysis of the various sub-types is still incomplete”. Note that we also included a more detailed comparison of previous studies and our own study in our revised manuscript as requested by the reviewers (see above and below).

To further characterize the different sub-types obtained from differentiation with RA and RA+PMA, we performed additional qPCR experiments targeting specific marker genes. These experiments allowed us to characterize the sub-type of neurons obtained from the two protocols (see new section in our revised manuscript).

Our statement regarding the differences of undifferentiated and differentiated cells was misleading. We wanted to emphasize that visible differences were not observed. We apologize for this confusion and modified the text accordingly. It now reads “Differences in the morphology of RA- and RA/PMA-differentiated cells were not recognised.”

2. The experimental setup is not totally clear to me (and maybe not straight forward). It is written, that undifferentiated cells were grown under standard conditions. Does this include that there were grown in 15% FBS? If this would be the case, it is not clear which effect the change of the serum concentration has on the cells and if the found differences at least in part might be due to the adaption to lower-serum conditions. This should at least be controlled e.g. by cells cultivated in exactly the same medium (2.5% FCS, solvent control?) with and without t RA / RA+PMA treatment. Furthermore, the differentiation protocols differ in length (6 days RA, 7 days RA+PMA). How can the authors be sure that the effect on the proteome really results from additional PMA treatment and not from the longer incubation time? What about the control? Have the control cells here been cultivated for 6 days or 7 days without passaging?

For clarity, we included a workflow in our revised manuscript (Figure 1, see also reviewer 1).

The major goal of our study was to compare two differentiated neuronal cell lines with their undifferentiated origins. For this, we followed commonly used protocols. We therefore did not vary these protocols and grew undifferentiated cells at standard conditions and differentiated cells under conditions typically employed for neuronal differentiation. This includes passaging of the cells during differentiation as the cells would otherwise overgrow. (Note that undifferentiated cells also undergo passaging.) During differentiation, cells are typically grown without FBS to subdue proliferation and enhance differentiation. A very similar protocol, including passaging and low-serum conditions, was employed in other proteomic studies (e.g. Zhang et. al.).

3. Line 68: "...and identify changes in the proteins' subcellular localization." As it is written here, I would expect a localization change of proteins within the cell. No experiment has been made to support that. Furthermore, if I understand the approach right, a very prominent abundance change of only one protein theoretically might be sufficient to give in sum a significant change. Would it be possible to either comment on this or to show some data e.g. the proportion of intensities of single proteins to address this?

We apologize for this confusion. The abundance corresponding to subcellular localization of the proteins changed. We clarified this in our manuscript.

4. The discussion is partly redundant and partly a repetition of the results. I'd suggest to remove the redundant parts and include instead e.g. a comparison to already published datasets e.g. Zhang et al. It would also be interesting which protein complexes have been described before and which might be novel.

We modified the discussion section and removed/shortened the repetitive parts. We also included our new findings (qPCR analyses) in the discussion as well as a comparison with other studies (see also comments of Reviewer 1).

5. Is there any reason that much more complexes were identified in differentiated cells or might this be an artifact e.g. through the higher amount of serum which has not been completely washed away before crosslinking? This might be checked by including also potential serum contaminants into the search. Has this been done for the crosslink data? In the protein abundance data, the last three replicates of both differentiated cells include about the double amount of potential serum contaminants. Is there a reason for this?

Serum components such as BSA are by default included in the database search ('contaminants'). We have checked the abundance of BSA in the different cell culture conditions and did not notice particularly high levels of BSA.

Minor remarks

1. line 30: "... reveal the presence of early differentiating neurons". I'd be cautious here; my feeling is that it quite difficult or impossible to show with the used methods that there are really neurons in the dish. If this claim is made, it should be proven at least by functional/physiological assays.

Our proteome analysis clearly identified neuronal marker proteins. Specifically, we identified marker proteins for early neuronal differentiation. We therefore conclude that neurons are obtained when following the described differentiation protocols. In addition, cell images show that neuronal networks are established. This is particularly apparent when comparing differentiated and undifferentiated cells. As part of the revision of our manuscript, we performed qPCR experiments and targeted specific marker genes. These experiments helped up to characterize the neuronal sub-types after differentiation.

2. Proteome analysis: Did I get it right, that only proteins were considered showing no missing value in all replicates of the respective comparisons? Were proteins identifications on the basis of only one peptide included?

The comparison between replicates allows assessing the reproducibility of the experiments. 3661 proteins were identified in all 18 replicates highlighting, as stated in the manuscript, a good reproducibility (as a quality measure). Note that not all of the 5909 total proteins are identified in all samples as the proteome changes during differentiation. Proteins that have not been identified in all but some of the 6 replicates of the same condition ('missing values') can be quantified through the MaxQuant algorithm. However, a pair-wise comparison of two conditions (as shown in Figure 3a) can only be performed for proteins that were quantified in both conditions. If a protein is missing in one condition (i.e. this protein was not identified in any of the 6 replicates of a condition), a fold-change cannot be calculated. We therefore distinguish between identification (based on selection of precursor ions during MS/MS analysis) and quantification (based on signal intensity in the MS spectra through MaxQuant even when corresponding precursors were not selected in all samples).

Yes, proteins that were identified by one peptide are included. Nowadays, mass spectrometers are characterized by high sensitive, high mass accuracy, high speed and high spectral quality, and data analysis tools improved over the last decades so that one-peptide hits are well-accepted in the proteomics community.

3. Line 128: What is meant with "significance threshold of 5%?"

A significance threshold of 5 % equals the alpha adjusted p-value 0.05 as given in the respective figure legend. A significance threshold of 5% is commonly applied in statistical analysis. Values above this significance threshold are not considered to be statistically relevant.

4. Legend Figure2: Should be "-log10"...

We apologize for this mistake. We corrected this typo.

5. Line 232: "We conclude that undifferentiated SH-SY5Y cells represent neuroblastoma cells...". That's not really a big surprise as this is already clear at least from the introduction...

We agree that this finding was not surprising, however, it is very important as undifferentiated cells are often used as neuronal model systems. The proteome and qPCR analysis clearly shows that undifferentiated cells do not show neuronal character and rather show cancer characteristics. Therefore, undifferentiated cells should not be used as neuronal model system.

6. Line 449: a protein:enzyme ratio 50:1 is mentioned. How has the protein concentration/amount been determined?

A protein:enzyme ratio of 50:1 is typically employed for tryptic hydrolysis. The amount of enzyme required was estimated based on the protein amount obtained after cell lysis. We have added this information to our revised manuscript.

REVIEWERS' COMMENTS:

Reviewer #2 (Remarks to the Author):

The authors provide a revised version of their manuscript. Changes include addition of phenotyping SH-SY5Y differentiation using qPCR, visualization of Histone crosslinks, addition of a visualization of the used workflow, shortening of the discussion and addition of a comparison to similar studies. A validation of found crosslinks/ proposed protein-protein interactions has not been performed.

The manuscript clearly profits from the changes and is now in a quite good shape. I'd like to thank the authors for implementing many of the suggested modifications and congratulated them to their excellent work.

Nevertheless, I'd like to stress some points which might be addressed:

1. Wording: As already mentioned, I'm not happy with the wording "neuron" which is used for differentiated cells at some points of the manuscript. A neuron is defined as electrically excitable cell. This has not been shown e.g. by electrophysiology. Therefore, I'd be more cautious here and would prefer a wording like "neuron-like".

2. Figure 4: Student's t-test have been used for statistical analysis. As three groups have been compared, an ANOVA with Post-Hoc tests should be more appropriate.

3. The authors argue that "Nowadays, mass spectrometers are characterized by high sensitive, high mass accuracy, high speed and high spectral quality, and data analysis tools improved over the last decades so that one-peptide hits are well-accepted in the proteomics community." I disagree, as the opposite might be true: The more peptides / proteins are identified the higher is the number of false positive identifications if e.g. an FDR base approach – as applied – is used. Please compare: <https://www.mcponline.org/mass-spec-guidelines> where the identification of proteins based on single peptide is discouraged. Nevertheless, the authors clearly stated how the data processing have been done, therefore, the readers have the possibility to judge on their own.

Reviewer #2 (Remarks to the Author):

The authors provide a revised version of their manuscript. Changes include addition of phenotyping SH-SY5Y differentiation using qPCR, visualization of Histone crosslinks, addition of a visualization of the used workflow, shortening of the discussion and addition of a comparison to similar studies. A validation of found crosslinks/ proposed protein-protein interactions has not been performed. The manuscript clearly profits from the changes and is now in a quite good shape. I'd like to thank the authors for implementing many of the suggested modifications and congratulated them to their excellent work.

Nevertheless, I'd like to stress some points which might be addressed:

We thank the reviewer for his positive comments. See below our detailed response.

1. Wording: As already mentioned, I'm not happy with the wording "neuron" which is used for differentiated cells at some points of the manuscript. A neuron is defined as electrically excitable cell. This has not been shown e.g. by electrophysiology. Therefore, I'd be more cautious here and would prefer a wording like "neuron-like".

We agree with the reviewer and revised the manuscript accordingly (see highlighted changes throughout the manuscript).

2. Figure 4: Student's t-test have been used for statistical analysis. As three groups have been compared, an ANOVA with Post-Hoc tests should be more appropriate.

Please note that we did not compare three groups. There are three groups analysed but the comparison is pair-wise. This pair-wise comparison is useful because statistical differences between undifferentiated and RA- or RA/PMA-differentiated cells (or RA- and RA/PMA-differentiated cells) are evaluated. This pair-wise comparison is commonly employed in these types of experiments.

3. The authors argue that "Nowadays, mass spectrometers are characterized by high sensitive, high mass accuracy, high speed and high spectral quality, and data analysis tools improved over the last decades so that one-peptide hits are well-accepted in the proteomics community." I disagree, as the opposite might be true: The more peptides / proteins are identified the higher is the number of false positive identifications if e.g. an FDR base approach – as applied – is used. Please compare: <https://www.mcponline.org/mass-spec-guidelines> where the identification of proteins based on single peptide is discouraged. Nevertheless, the authors clearly stated how the data processing have been done, therefore, the readers have the possibility to judge on their own.

We agree with the reviewer that these "single peptide hits" are discouraged in the given guidelines, however the guidelines also state that "if included, the ability to view annotated spectra for these identifications must be made available." According to the guidelines, annotated spectra are provided in different ways including upload to a data repository. According to the guidelines of Communications Biology, we uploaded all data and results files. The reader can access these data and review single peptide hits.